# Quantitative modeling of the emergence of macroscopic grid-like representations

**Ikhwan Bin Khalid**[1,2,3†], **Eric T Reifenstein**[1,2,4†], **Naomi Auer**[2], **Lukas Kunz**[5*‡§], **Richard Kempter**[1,2,3*‡#]

[1]Bernstein Center for Computational Neuroscience Berlin, Berlin, Germany; [2]Institute for Theoretical Biology, Department of Biology, Humboldt-Universität zu Berlin, Berlin, Germany; [3]Einstein Center for Neurosciences Berlin, Berlin, Germany; [4]Department of Mathematics and Computer Science, Freie Universität Berlin, Berlin, Germany; [5]Department of Epileptology, University Hospital Bonn, Bonn, Germany

**\*For correspondence:**
lukas.kunz@ukbonn.de (LK);
r.kempter@biologie.hu-berlin.de (RK)

†These authors contributed equally to this work
‡These authors also contributed equally to this work

**Present address:** §Department of Epileptology, University Hospital Bonn, Bonn, Germany; #Department of Biology, Institute for Theoretical Biology, Humboldt-Universität zu Berlin, Berlin, Germany

**Competing interest:** The authors declare that no competing interests exist.

**Abstract** When subjects navigate through spatial environments, grid cells exhibit firing fields that are arranged in a triangular grid pattern. Direct recordings of grid cells from the human brain are rare. Hence, functional magnetic resonance imaging (fMRI) studies proposed an indirect measure of entorhinal grid-cell activity, quantified as hexadirectional modulation of fMRI activity as a function of the subject's movement direction. However, it remains unclear how the activity of a population of grid cells may exhibit hexadirectional modulation. Here, we use numerical simulations and analytical calculations to suggest that this hexadirectional modulation is best explained by head-direction tuning aligned to the grid axes, whereas it is not clearly supported by a bias of grid cells toward a particular phase offset. Firing-rate adaptation can result in hexadirectional modulation, but the available cellular data is insufficient to clearly support or refute this option. The magnitude of hexadirectional modulation furthermore depends considerably on the subject's navigation pattern, indicating that future fMRI studies could be designed to test which hypothesis most likely accounts for the fMRI measure of grid cells. Our findings also underline the importance of quantifying the properties of human grid cells to further elucidate how hexadirectional modulations of fMRI activity may emerge.

## Editor's evaluation

This computational work represents a valuable and long overdue assessment of the potential mechanisms associating patterns of activity of entorhinal grid cells, recorded mostly in rodents, with the population property of hexasymmetry detected in non-invasive human studies. The methodic comparison of alternative hypotheses is compelling, and the conclusions are important for the future design of experiments assessing the neural correlates of human navigation across physical, virtual, or conceptual spaces.

## Introduction

The neural basis of spatial navigation comprises multiple specialized cell types such as place cells (*O'Keefe and Dostrovsky, 1971*), head-direction (HD) cells (*Taube et al., 1990*), and grid cells (*Hafting et al., 2005*), whose activity profiles result from intricate mechanisms of microcircuits in the medial temporal lobes (*Tukker et al., 2022*). Grid cells are neurons that activate whenever an animal or human traverses the vertices of a triangular grid tiling the entire environment into equilateral triangles (*Hafting et al., 2005*; *Jacobs et al., 2013*). Grid cells may allow the navigating organism to perform vector computations and may thus constitute an essential neural substrate for different types

of spatial navigation including path integration, though their exact functional role still remains unclear (*Stemmler et al., 2015*; *Bush et al., 2015*; *Moser et al., 2017*; *Stangl et al., 2018*; *Gil et al., 2018*; *Banino et al., 2018*; *Bierbrauer et al., 2020*; *Ginosar et al., 2023*).

In rodents, grid cells can be recorded using electrodes inserted into the medial entorhinal cortex (EC). In humans, measuring grid cells using invasive methods is only rarely possible, e.g., by recording single-neuron activity in epilepsy patients who are neurosurgically implanted with intracranial depth electrodes (*Jacobs et al., 2013*; *Nadasdy et al., 2017*). Hence, to enable the detection of grid cells in healthy humans, a functional magnetic resonance imaging (fMRI) method has been developed that tests for a hexadirectional modulation of the blood-oxygen-level-dependent signal as a function of the subject's movement direction through a virtual environment (*Doeller et al., 2010*). We here refer to this phenomenon of a hexadirectional modulation of the fMRI signal as 'macroscopic grid-like representations', which has been replicated repeatedly in recent years (e.g. *Kunz et al., 2015*; *Bellmund et al., 2016*; *Horner et al., 2016*; *Constantinescu et al., 2016*; *Bierbrauer et al., 2020*). The mechanisms underlying the emergence of such macroscopic grid-like representations remain still unclear, however.

To provide possible explanations for the emergence of macroscopic grid-like representations, previous studies presented several qualitatively different hypotheses on how the activity of single grid cells translates into a macroscopically visible hexadirectional fMRI signal (*Doeller et al., 2010*; *Kunz et al., 2019*). Three main hypotheses have been developed: (i) the 'conjunctive grid by HD cell hypothesis'; (ii) the 'repetition suppression hypothesis'; and (iii) the 'structure-function mapping hypothesis' (*Figure 1*).

The conjunctive grid by HD cell hypothesis rests on two key findings: first, the existence of conjunctive grid by HD cells, which were found in the deeper layers of the entorhinal cortex and in pre- and parasubiculum (*Sargolini et al., 2006*); second, the observation that the directional tuning of grid cells within the entorhinal cortex is aligned with the grid axes (*Doeller et al., 2010*), though further studies are needed to corroborate this observation. Assuming that the directional tuning width of these conjunctive grid by HD cells is not too broad, movements aligned with the grid axes (as compared to misaligned movements) result in increased spiking activity of the conjunctive grid by HD cell population. Given some correlation between population spiking activity and the fMRI signal, this systematic difference in the firing of conjunctive grid by HD cells when moving aligned versus misaligned with the grid axes may thus cause a macroscopically visible fMRI signal with hexadirectional modulation (*Figure 1B*). We note that the conjunctive grid by HD cell hypothesis does not necessarily depend on an alignment between the preferred head directions with the grid axes.

The repetition suppression hypothesis (*Figure 1C*) is based on the assumption that the phenomenon of repetition suppression—i.e., neural activity being reduced for repeated stimuli (*Grill-Spector et al., 2006*)—also applies to grid cells (*Doeller et al., 2010*; *Killian et al., 2012*). Critical to this hypothesis is that relatively fewer different grid cells are activated more often during movements aligned with the grid axes, and relatively more different grid cells are activated less often during misaligned movements. Due to this systematic difference in how many grid cells are activated how often, a higher degree of repetition suppression at the level of spiking activity or the fMRI signal (i.e. fMRI adaptation) during aligned movements as compared with misaligned movements can emerge, again resulting in a hexadirectional modulation of fMRI activity as a function of the subject's movement direction through the spatial environment.

Regarding the structure-function mapping hypothesis (*Figure 1D*), studies in rodents have demonstrated that the firing fields of anatomically adjacent grid cells do not only have similar spacing and orientation (*Stensola et al., 2012*), but also a similar grid phase offset to a reference location (*Heys et al., 2014*; *Gu et al., 2018*). Studies in rodents observed that grid cells also cluster anatomically (*Obenhaus et al., 2022*; *Naumann et al., 2018*), which may lead to stronger grid-like representations in some voxels (with an increased percentage of grid cells) than in others (with a decreased percentage of grid cells). Therefore, recordings from a small area of the EC (e.g. a sufficiently small voxel of an fMRI scan) may sample grid cells with similar firing fields, which basically behave similarly. It has been suggested that such a grid cell population might show higher average firing rates during aligned movements (because more firing fields are traversed) versus misaligned movements, again resulting in macroscopically visible grid-like representations (*Kunz et al., 2019*).

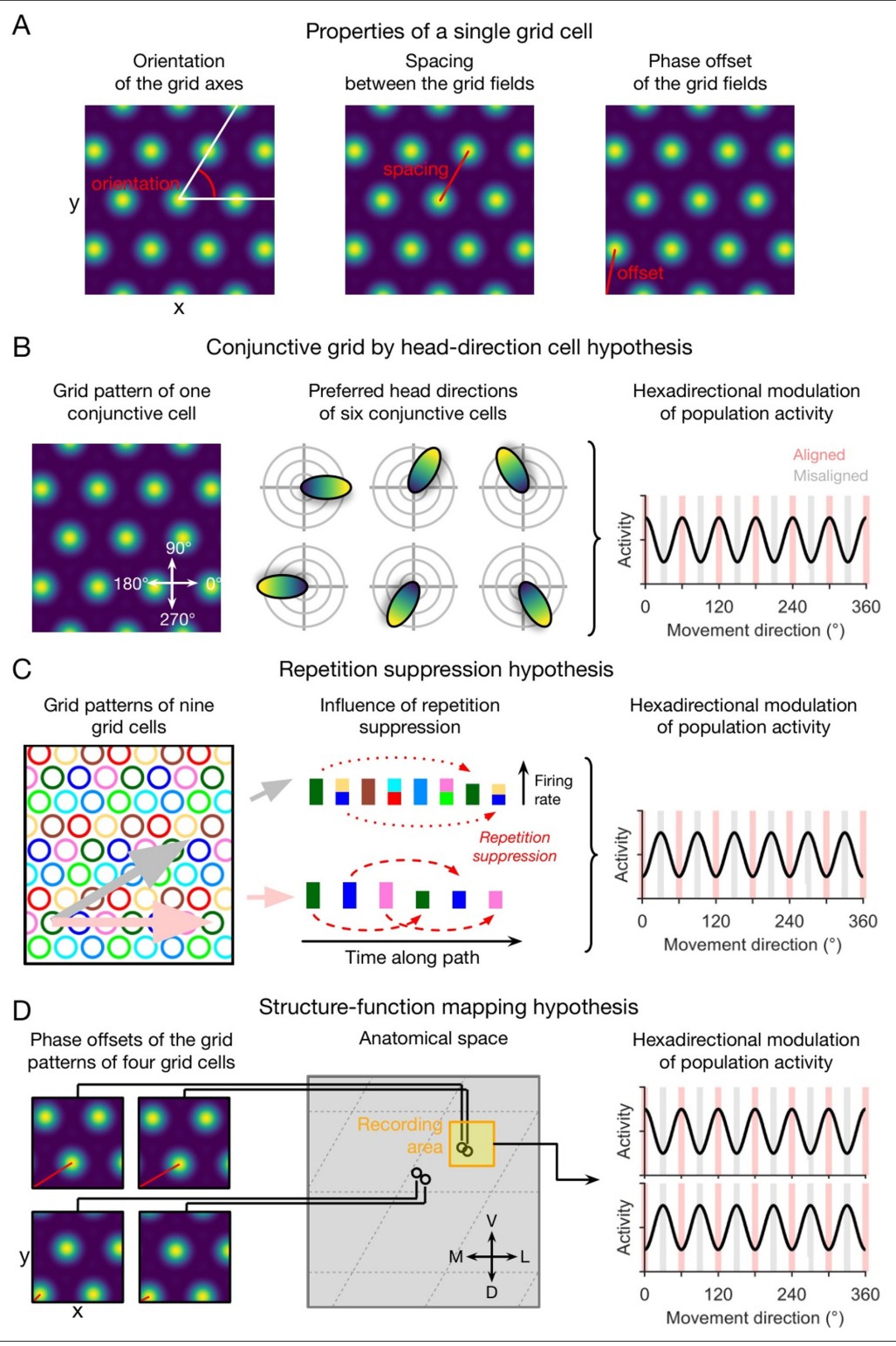

**Figure 1.** Qualitative hypotheses on the emergence of macroscopic grid-like representations in the human entorhinal cortex. (**A**) Grid-cell properties comprise grid orientation, grid spacing, and grid phase offset. (**B**) Conjunctive grid by head-direction cell hypothesis. Macroscopic grid-like representations (right) may emerge from the firing of conjunctive grid by head-direction cells (left and middle) that exhibit increased firing when the subject moves aligned as compared to misaligned with the grid axes (right) (*Doeller et al., 2010*). (**C**) Repetition suppression hypothesis. In a grid cell population with similar grid orientations and grid spacings but with

*Figure 1 continued on next page*

*Figure 1 continued*

distributed phases (left; colored circles represent firing fields of different grid cells), aligned movements (horizontal pink arrow) lead to more frequent activation (shorter distance between firing fields) of a smaller number of different grid cells, whereas misaligned movements (diagonal gray arrow) lead to less frequent activation (larger distance between firing fields) of a higher number of different grid cells (*Doeller et al., 2010*). Thus, aligned movements may lead to more pronounced repetition suppression as compared to misaligned movements (middle), resulting in a hexadirectional modulation of population spiking activity and thus in the emergence of grid-like representations (right). (**D**) Structure-function mapping hypothesis. Because anatomically adjacent grid cells exhibit similar grid phase offsets (in addition to similar grid orientations and grid spacings) (*Gu et al., 2018*), recordings from a limited number of grid cells with a non-random distribution of phase offsets may lead to macroscopic grid-like representations. The left panel shows the grid phase offset of four different grid cells, whose anatomical locations are illustrated in the middle panel. Depending on the subject's starting location relative to the phase offset of the grid fields, movements aligned or misaligned with the grid axes lead to higher sum grid cell activity as compared to misaligned or aligned movements (right panel). Furthermore, the orientation of hexadirectional modulation may shift when recording from neighboring voxels in anatomical space due to a shift in the clustered phase offsets. D, dorsal; L, lateral; M, medial; V, ventral. Figure 1 has been adapted from Figure 3 from *Kunz et al., 2019*.

The online version of this article includes the following figure supplement(s) for figure 1:

**Figure supplement 1.** Effect of tortuosity on the hexasymmetry for the repetition suppression hypothesis with random walks ($\Delta t = 0.01$ s, $T = 9000$ s, $\tau_r = 3$ s, $w_r = 1$).

**Figure supplement 2.** Effect of size and shape of finite environments on hexasymmetry.

**Figure supplement 3.** Effect of rotation of finite environments on hexasymmetry.

**Figure supplement 4.** Hexasymmetry of random-walk trajectories.

In this study, we aimed at quantitatively evaluating the three hypotheses on the emergence of macroscopic grid-like representations using a modeling approach. Our results show that all three hypotheses can result in macroscopic grid-like representations under ideal and specific conditions, but that the strength of the hexadirectional modulation varies by orders of magnitude. Key findings are also that the subjects' type of navigation paths through the spatial environments and the exact biological characteristics of grid cells determine to what extent a given hypothesis can explain a hexadirectional population signal in the EC. In this way, our results help understand how grid cells may have a specific correlate in fMRI, make predictions on how future fMRI studies could establish evidence in favor of or against one of the three hypotheses, and suggest that the biological properties of grid cells in humans should be investigated in greater detail in order to support or weaken the plausibility of either of the three hypotheses.

## Results

In what follows, we first describe the navigation strategies that we use in our model, then define a new measure to quantify neural hexasymmetry and path hexasymmetry, and finally calculate neural hexasymmetries for the three hypotheses, which we model for idealized as well as more realistic choices of parameters.

### Navigation strategies

To evaluate the different hypotheses on the emergence of grid-like representations, we considered three different types of navigation trajectories: 'star-like walks', 'piecewise linear walks', and 'random walks'. We opted for this approach in order to test whether a subject's navigation pattern—which in itself comes with a certain degree of hexasymmetry ('path hexasymmetry')—influences the emergence of hexadirectional sum signals of neuronal activity.

During each path segment of star-like walks, the simulated agent started from the same (x/y)-coordinate and navigated along 1 of 360 predefined allocentric navigation directions (0° to 359° in steps of 1°; *Figure 2B*, left). This ensured that the navigation trajectory itself exhibited a hexasymmetry that was essentially zero. Each path had a length of 300 cm, which was 10 times the grid scale (see *Table 1* for a summary of all model parameters). After each path segment was traversed, the agent was 'teleported' back to the initial (x/y)-coordinate and completed the next path segment. For

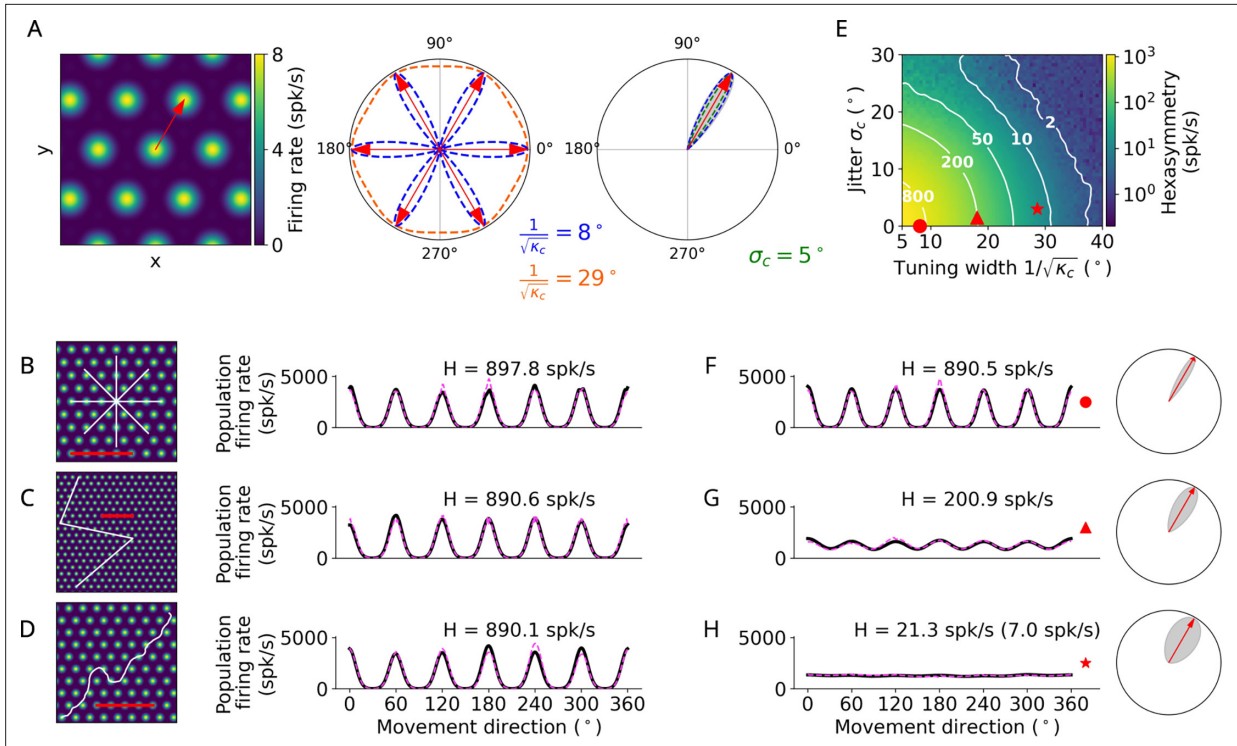

**Figure 2.** Conjunctive grid by head-direction (HD) cell hypothesis. (**A**) Left: The preferred head direction (red arrow) of a single conjunctive cell is aligned to one of the grid axes (*Doeller et al., 2010*). Two factors add noise to this relation: the HD tuning has a certain width ($1/\sqrt{\kappa_c}$) and the alignment of grid orientation to HD tuning angle is jittered ($\sigma_c$). Middle: Distribution of the tuning width $1/\sqrt{\kappa_c}$ around all possible grid axes for two example values. Right: Convolution (gray) between the distributions of the jitter $\sigma_c = 5°$ and tuning width $1/\sqrt{\kappa_c} \approx 8.1°$ ($\kappa_c = 50 \text{ rad}^{-2}$) to obtain the effective HD tuning of a single conjunctive cell around one grid axis. (**B–D**) Simulation of the conjunctive hypothesis using 'ideal' parameters of $\kappa_c = 50 \text{ rad}^{-2}$ and $\sigma_c = 0°$. The scale bars (red) represent a distance of 120 cm. (**B**) Left: Illustration of a star-like walk (path segments are cut for illustration purposes), overlaid onto the firing pattern of a single grid cell. Right: Population firing rate as a function of the subject's movement direction (which is identical with heading direction in our simulations) for star-like runs with mean firing rate $\tilde{A}_0 = 1279.7$ spk/s (for 1024 cells) and path hexasymmetry $|\tilde{T}_6| < 10^{-10}$; see Methods for definitions of $\tilde{A}_0$ and $\tilde{T}_6$. (**C**) Left: Illustration of a piecewise linear walk (cut for illustration purposes), overlaid onto the firing pattern of a single grid cell. Right: Population firing rate as a function of movement direction for piecewise linear walks with $\tilde{A}_0 = 1279.3$ spk/s and $|\tilde{T}_6| < 10^{-10}$. (**D**) Left: Illustration of a random walk (cut for illustration purposes), overlaid onto the firing pattern of a single grid cell. Right: Population firing rate as a function of movement direction for random walks with $\tilde{A}_0 = 1281.0$ spk/s and $|\tilde{T}_6| = 6.7 \cdot 10^{-3}$. (**E**) Hexasymmetry (color coded) as a function of HD tuning width and alignment jitter for star-like walk trajectories at an offset of $(0, 0)$. Higher hexasymmetry values are achieved for stronger HD tuning and tighter alignment of the preferred head directions to the grid axes. The red symbols correspond to the three parameter combinations used in subplots (**B–D**) and (**F–H**) for further illustration. Large tuning widths ($\kappa_c \to 0$) correspond to cosine tuning for which the hexasymmetry approaches 0. (**F**) Population firing rate as a function of movement direction for a random-walk trajectory with jitter $\sigma_c = 0$ and concentration parameter $\kappa_c = 50 \text{ rad}^{-2}$ (tuning width $\approx 8.1°$). (**G**) Population firing rate as a function of movement direction for a random walk with jitter $\sigma_c = 1.5°$ and concentration parameter $\kappa_c = 10 \text{ rad}^{-2}$ (tuning width $\approx 18.1°$). (**H**) Population firing rate as a function of movement direction for a random walk with jitter $\sigma_c = 3°$ and concentration parameter $\kappa_c = 4 \text{ rad}^{-2}$ (tuning width $\approx 28.6°$). The hexasymmetry for the case of $p_c = 33\%$ is stated in brackets. All other simulations presented in this figure use $p_c = 100\%$ conjunctive ($N = 1024$) cells, which is higher than in empirical studies (*Sargolini et al., 2006*; *Boccara et al., 2010*). In subplots (**B–D**) and (**F–H**), the black solid lines and magenta dashed lines represent the results from the numerical simulations of *Equation 8* and the analytical derivation in *Equation 40*, respectively. The radial subplots in (**F–H**), right, illustrate the effective HD tuning width around a single grid axis analogous to A, right. H, neural hexasymmetry; spk/s, spikes per second.

The online version of this article includes the following figure supplement(s) for figure 2:

**Figure supplement 1.** Comparison between the 'ideal' tuning width and 'realistic' tuning widths found in existing literature for the conjunctive grid by head-direction (HD) cell hypothesis.

**Figure supplement 2.** Dependence of the hexasymmetry on the proportion of conjunctive grid by head-direction cells.

real-world experiments, this type of navigation including teleportation is unusual, but it can be implemented in virtual-reality experiments (*Vass et al., 2016*; *Deuker et al., 2016*).

During piecewise linear walks, the subject also completed 360 path segments of 300 cm length along the same 360 predefined allocentric navigation directions, as in the star-like walks. In this case,

**Table 1.** Parameters: descriptions and values.

For a more detailed motivation of the values used, see section 'Parameter estimation'.

| Parameter | Description | Values (unless varied) or range |
|---|---|---|
| **Trajectories** | | |
| $\Delta t$ | Simulation time step | 0.01 s |
| $T$ | Simulated duration | 9000 s |
| $v$ | Movement speed | 10 cm/s |
| $\sigma_\theta$ | Movement tortuosity | 0.5 rad/s$^{1/2}$ |
| $r_{max}$ | Length of a linear path in the star-like run | 300 cm |
| $N_\theta$ | Number of angles to sample in the star-like run | 360 |
| **Grid cells** | | |
| $N$ | Number of grid cells in a voxel | 1024 |
| $s$ | Grid scale | 30 cm |
| $\gamma$ | Grid orientation | 0° |
| $A_{max}$ | Maximum firing rate for one grid cell | 8 spk/s |
| $(x_{off}, y_{off})$ | Grid phase (two-dimensional) | ([0,1], [0,1]) |
| **Conjunctive grid by head-direction cell hypothesis** | | |
| $\mu_c$ | Preferred head direction | $[0, 2\pi)$ (multiples of 60° for $\sigma_c = 0$) |
| $\kappa_c$ | Concentration parameter for direction tuning for the ideal and 'realistic' cases | {50, 4} rad$^{-2}$ |
| $\sigma_c$ | Alignment jitter of direction tuning to grid axis for the ideal and 'realistic'* cases | {0, 3}° |
| $p_c$ | Fraction of conjunctive cells in a population for the ideal and 'realistic'* cases | {1, 1/3} |
| **Repetition suppression hypothesis** | | |
| $\tau_r$ | Adaptation time constant for the ideal and 'weaker' cases | {3, 1.5} s |
| $w_r$ | Adaptation weight for the ideal and 'weaker' cases | {1, 0.5} |
| **Structure-function mapping hypothesis** | | |
| $\mu_s$ | Central phase of cluster | (0, 0) |
| $\kappa_s$ | Concentration parameter for clustering for the ideal and 'realistic'[†] cases | {10, 0.1} |

*__Boccara et al., 2010__; __Sargolini et al., 2006__.
[†]__Gu et al., 2018__.

however, the path segments were 'unwrapped' such that the starting location of a path segment was identical with the end location of the preceding path segment (__Figure 2C__, left). The sequence of allocentric navigation directions was randomly chosen. As for star-like walks, piecewise linear walks do not exhibit hexasymmetry a priori.

For random walks, we modeled navigation trajectories following __Kropff and Treves, 2008__; __Si et al., 2012__; __D'Albis and Kempter, 2017__ (for details, see Methods around __Equation 1__), which allowed us to vary the tortuosity of the paths. For a certain value of the tortuosity parameter ($\sigma_\theta = 0.5$ rad/s$^{1/2}$) and a time step $\Delta t = 0.01$ s, this led to navigation paths that we considered similar to those seen in rodent studies (__Figure 2D__, left; __Figure 1—figure supplement 1E__). Tortuosity describes how convoluted a navigation path is (the higher the tortuosity, the more convoluted the path). A value of 0.5 rad/s$^{1/2}$ means that in 1 s of time the standard deviation of movement direction change is about 0.5 rad = 29° (see __Figure 1—figure supplement 1E__, and __Equation 42__ for $m\Delta t = 1$ s).

Apart from random walks in basically infinite environments, we also simulated random walks in finite enclosures (circles and squares) with different sizes and orientations; we found that these restrictions had a negligible effect on path hexasymmetry (*Figure 1—figure supplement 2C* and *Figure 1—figure supplement 3C*; for a definition of path hexasymmetry, see the paragraph after the next one). Because the allocentric navigation directions are not predefined for random walks, they exhibit varying degrees of path hexasymmetry. The longer the simulated random walks, the smaller the path hexasymmetry (*Figure 1—figure supplement 4*). We simulated random walks with a total length of typically 900 m ($M = 9 \cdot 10^5$ steps).

As we describe below, the emergence of grid-like representations based on the conjunctive hypothesis is robust against the specific type of navigation strategy, whereas the other two hypotheses are sensitive to particular navigation strategies. Future studies on hexadirectional signals should thus consider the kind of navigation paths subjects will use during a given task.

## Quantifying neural hexasymmetry generated by the three hypotheses

To test how the activity of grid cells could give rise to hexasymmetry of a macroscopic signal, we used a firing-rate model of grid cell activity (*Equation 2*). Furthermore, we developed a new measure $H$ to quantify neural hexasymmetry (see Methods, *Equation 12*), which is the magnitude of the hexadirectional modulation of the summed activity of many cells. The hexasymmetry $H$ has a value $H = 0$ if there is no hexadirectional modulation, e.g., if the population firing rate does not depend on the movement direction. Conversely, if the population firing rate has a hexadirectional sinusoidal modulation, half of its amplitude (in units of the firing rate) equals the value of the hexasymmetry $H$. Using the same approach, we quantified the hexasymmetry of a trajectory and called this path hexasymmetry (see Methods after *Equation 14*).

### Conjunctive grid by HD cell hypothesis

The conjunctive grid by HD cell hypothesis (*Doeller et al., 2010*) suggests that hexadirectional activity in the EC emerges due to grid cells whose firing rate is additionally modulated by head direction, whereby the preferred head direction is closely aligned with one of the grid axes (*Figure 2A*). By modulating the activity of individual grid cells with an HD tuning term aligned with the grid axes (Methods, *Equation 3*), our simulations indeed showed that these properties resulted in a clear hexadirectional modulation of sum grid-cell activity (*Figure 2B–D*). When considering different types of navigation trajectories, we found that they led to similar distributions of sum grid-cell activity as a function of movement direction and, accordingly, to similar hexasymmetry values (*Figure 2B–D*). In all three cases, the directions of maximal activity were aligned with the grid axes.

These results were obtained using ideal values for the preciseness of the HD tuning (i.e. large concentration parameter of HD tuning, $\kappa_c$) and for the alignment of the preferred head directions to the grid axes (i.e. small alignment jitter of the HD tuning to the grid axes, $\sigma_c$; *Figure 2A*). We were thus curious how the hexasymmetry changed when using a wide range of parameter values that would also include biologically plausible values. We varied $\kappa_c$ between values corresponding to narrow tuning widths ($\kappa_c = 50$ rad$^{-2}$, which corresponds to an angular variability of $1/\sqrt{\kappa_c} \approx 8°$) and wide tuning widths ($\kappa_c = 4$ rad$^{-2}$, i.e. an angular variability of approximately $29°$), and we varied $\sigma_c$ between values of no jitter ($\sigma_c = 0$) and significant jitter ($\sigma_c = 3°$); see also *Table 1* for a summary of parameters. We found that a combination of narrow HD tuning widths and no jitter resulted in the largest hexasymmetry $H$ (*Figure 2E–H*), while wider tuning widths with non-zero jitter resulted in smaller values for the hexasymmetry (for larger tuning widths and a comparison to experimental values, see *Figure 2—figure supplement 1*).

### Repetition suppression hypothesis

Next, we performed simulations to understand whether the repetition suppression hypothesis (*Doeller et al., 2010*) results in a hexadirectional modulation of population grid-cell activity. This hypothesis proposes that grid-cell activity is subject to firing-rate adaptation and thus leads to reduced grid-cell activity when moving along the grid axes as compared to when moving along other directions than the grid axes (*Figure 3A*). This difference is due to the fact that when the subject moves along the grid axes the grid fields of fewer grid cells are traversed relatively more often (strong repetition

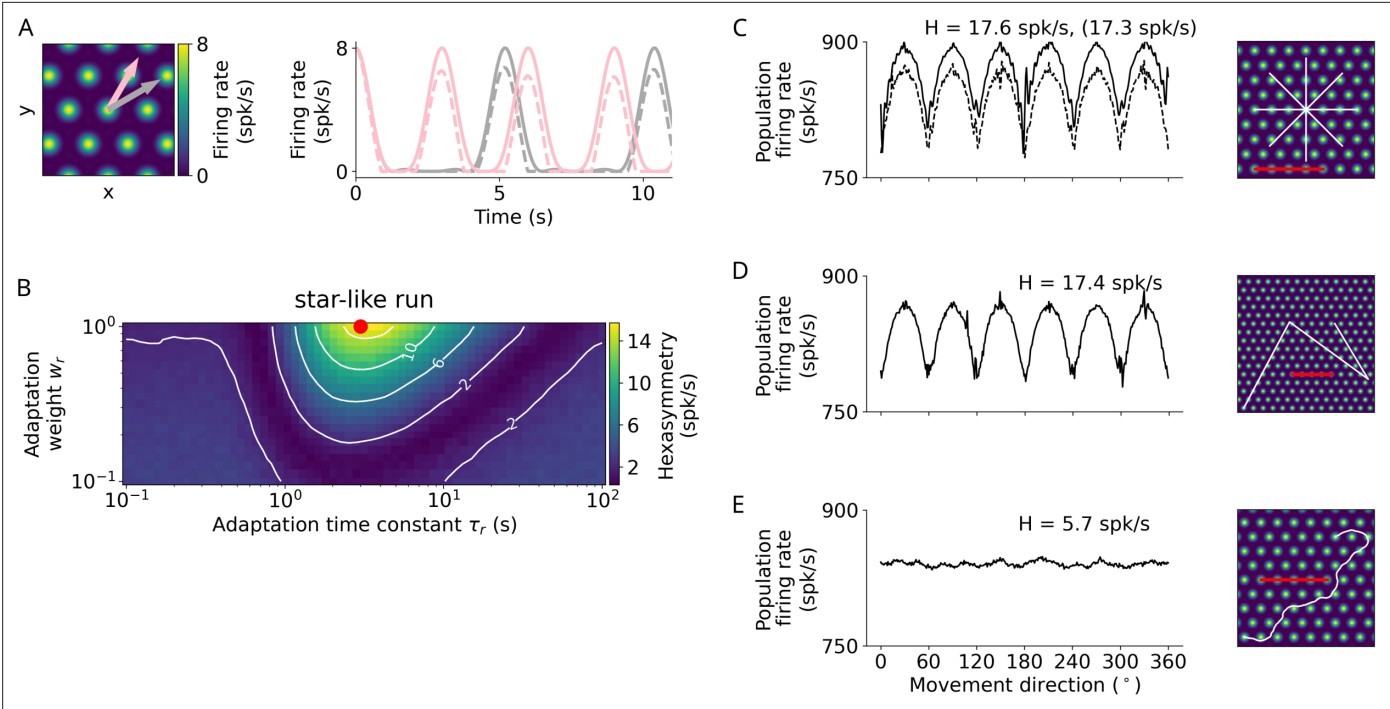

**Figure 3.** Repetition suppression hypothesis. (**A**) Left: Tuning of an example grid cell with aligned (pink arrow) and misaligned (gray arrow) movement directions. Right: Examples of firing-rate adaptation (dashed lines, adaptation weight $w_r = 1$ and time constant $\tau_r = 3$ s) for an aligned run (pink) and a misaligned run (gray). More firing-rate adaptation (i.e. stronger repetition suppression) occurs along the aligned run compared to the misaligned run (attenuation of peaks: 24% and 18% respectively). For both runs, firing rates are reduced compared to the case without adaptation (solid lines). (**B**) Simulations of hexasymmetry as a function of the weight ($w_r$) and time constant ($\tau_r$) of adaptation for star-like runs. The red dot marks the optimal parameters ($w_r = 1$ and $\tau_r = 3$ s) used in panels A and C–E. (**C**) Population firing rate as a function of the subject's movement direction for a star-like run at an offset of $(0, 0)$. The solid line represents firing rates for single runs where adaptation does not carry over when sampling different movement directions, i.e., the 'teleportation' between path segments resets the repetition suppression effects, and movement directions are sampled consecutively from $0°$ to $359°$ in steps of $1°$ (mean firing rate $\tilde{A}_0 = 866.4$ spk/s, path hexasymmetry $|\tilde{T}_6| < 10^{-10}$). The dashed line represents firing rates for single runs with adaptation carry-over and randomly sampled movement directions without replacement ($\tilde{A}_0 = 839.8$ spk/s, path hexasymmetry $|\tilde{T}_6| < 10^{-10}$, the corresponding hexasymmetry is shown in brackets). (**D**) Population firing rate as a function of movement direction for a piecewise linear walk ($\tilde{A}_0 = 839.0$ spk/s, $|\tilde{T}_6| < 10^{-10}$). (**E**) Population firing rate as a function of movement direction for a random walk ($\tilde{A}_0 = 839.7$ spk/s, $|\tilde{T}_6| = 6.7 \cdot 10^{-3}$). spk/s, spikes per second. For all repetition suppression simulations, the grid phase offsets of 1024 grid cells were sampled randomly from a uniform distribution across the unit rhombus, and the hexasymmetries were averaged over 20 realizations. The scale bars (red) in panels C–E represent a distance of 120 cm.

suppression), whereas when moving not along the grid axes the grid fields of more grid cells are traversed relatively less often (weak repetition suppression) (**Doeller et al., 2010**).

In our model, the repetition suppression hypothesis depends on two adaptation parameters: the adaptation time constant $\tau_r$ and the adaptation weight $w_r$ (**Equation 5**). We explored a large range of adaptation time constants and found that the time constant that leads to the largest hexasymmetry is roughly the subject's speed $v$ divided by the grid scale $s$ (**Figure 3B**). We constrained values of the adaptation weight to the full range of reasonable values ($0 < w_r \leq 1$) and found that the larger the value of the adaptation weight the larger is the hexasymmetry (**Figure 3B**). When examining how the different types of navigation trajectories affected hexadirectional modulations based on repetition suppression, we found that star-like and piecewise linear walks resulted in clear and significant hexasymmetry values (**Figure 3C and D**), which is driven by the long linear segments in these trajectory types. In contrast, random walks did not result in a significant hexadirectional modulation of sum grid-cell activity because the tortuosity of the random walk that we typically used ($\sigma_\theta = 0.5$ rad/s$^{1/2}$) is too large (i.e. trajectories turn too much between two adjacent firing fields of a grid cell) to be able to exploit the movement-direction dependence of repetition suppression (**Figure 3E**). Examining in more detail which tortuosity values would still lead to some hexadirectional modulation due to repetition suppression, we found that $\sigma_\theta \lesssim 0.25$ rad/s$^{1/2}$ was the upper bound. For smaller values of the

tortuosity parameter trajectories are straight enough to allow for a hexadirectional modulation of sum grid-cell activity (*Figure 1—figure supplement 1*).

A notable difference of the repetition suppression hypothesis compared to the conjunctive grid by HD cell hypothesis is that the apparent preferred grid orientation (i.e. the movement directions resulting in the highest sum grid-cell activity) is shifted by $30°$ and is thus exactly misaligned with the grid axes of the individual grid cells (*Figure 3C and D*). This is due to the fact that the adaptation mechanism suppresses grid-cell activity more strongly when moving aligned with a grid axis as compared to when moving misaligned with a grid axis (*Figure 3A*).

## Structure-function mapping hypothesis

We next investigated the structure-function mapping hypothesis, according to which a hexadirectional modulation of EC activity emerges in situations when a population of grid cells is recorded whose grid phase offsets are biased toward a particular offset (*Kunz et al., 2019*). In the ideal case, all grid phase offsets are identical, and thus all grid cells behave like a single grid cell. This hypothesis is called 'structure-function mapping hypothesis' because of a direct mapping between the anatomical locations of the grid cells in the EC and their functional firing fields in space.

We found indeed that highly clustered grid phase offsets ('ideal' concentration parameter for clustering $\kappa_s = 10$) resulted in significant hexadirectional modulations of sum grid-cell activity when the subject performed star-like walks starting at a phase offset of $(0, 0)$, i.e., the center of the cluster of firing fields of grid cells (*Figure 4A*). Interestingly, the hexasymmetry values during star-like walks were strongly dependent on the subject's starting location relative to the locations of the grid fields as only particular starting phases (within the unit rhombus of grid phase offsets) such as $(0, 0)$ or $(0.3, 0.3)$ led to clear hexasymmetry whereas others, e.g., $(0.6, 0)$, did not (*Figure 4D*, left). Additionally, the 'apparent preferred grid orientation' (i.e. the movement directions associated with the highest sum grid-cell activity) was shifted by $30°$ for certain offsets in the unit rhombus illustrating the subject's starting locations relative to the firing-field locations (*Figure 4D*, right). It was furthermore notable that the summed grid-cell activity as a function of movement direction exhibited relatively sharp peaks at multiples of $60°$ with additional small peaks in between (*Figure 4A*, right). This pattern is clearly distinct from the more sinusoidal modulation of sum grid-cell activity resulting from the conjunctive grid by HD cell hypothesis (*Figure 2*) and the more full-wave rectified sinusoidal modulation resulting from the repetition suppression hypothesis (*Figure 3*).

When examining the structure-function mapping hypothesis for piecewise linear walks and random walks, hexasymmetry appeared to be considerably lower as compared to simulations with star-like walks (*Figure 4B and C*). For piecewise linear walks (but not for random walks), population grid-cell activity as a function of movement direction again exhibited sharp peaks at multiples of $60°$, in both positive and negative directions (*Figure 4B*). For both piecewise linear walks and random walks, the hexasymmetry values did not show a systematic dependency on the starting location of the subject's navigation trajectory relative to the locations of the grid fields (*Figure 4E and F*, left), which is due to the fact that these navigation trajectories randomize the starting locations of all path segments. Accordingly, the apparent preferred grid orientations varied randomly as a function of the starting location of the very first path segment and did not exhibit systematic shifts (*Figure 4E and F*, right), which is in contrast to the clear shifts in the apparent preferred grid orientations for star-like walks (*Figure 4D*).

In the above-mentioned simulations for the structure-function mapping hypothesis, we chose high values for the clustering of the grid phase offsets from all grid cells. Specifically, we set the clustering concentration parameter to $\kappa_s = 10$, due to which the centers of the firing fields of different grid cells were very close to each other (insets in *Figure 4A–C*, right). However, previous empirical studies (*Gu et al., 2018*; *Heys et al., 2014*) suggest that the clustering is much weaker. We thus performed numerical simulations to obtain a more realistic estimation of the clustering concentration parameter, $\kappa_s$. We observed that our numerically determined clustering concentration parameter that matched the empirical clustering was in fact closer to randomly distributed grid phase offsets ($\kappa_s < 0.1$; *Figure 4G–S*) than to strong, ideal clustering ($\kappa_s = 10$). Specifically, we implemented a 'simple' clustering using a Gaussian kernel that led to a clustering concentration parameter $\kappa_s \approx 0.014$ (*Figure 4M*). The clustering concentration parameter remained low when we added an empirically guided 'higher-order' spatial clustering of grid phase offsets by adding some longer-range spatial autocorrelations to

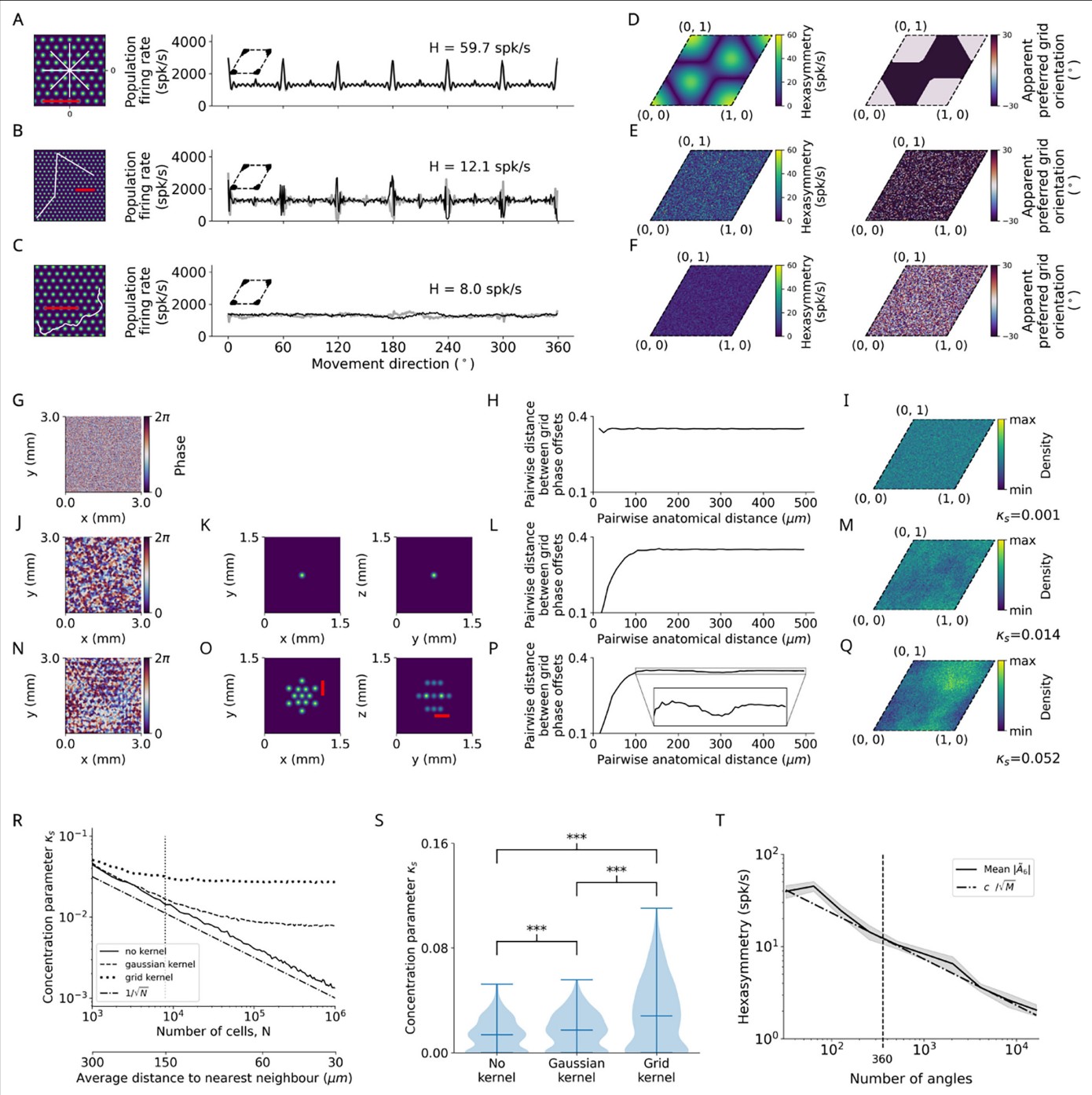

**Figure 4.** Structure-function mapping hypothesis. (**A–C**) Left: Short example trajectories (white) overlaid onto the firing-rate pattern of an example grid cell (colored). Shown trajectories are for illustration purposes only, and do not reflect the full length of the simulation. The scale bars (red) represent a distance of 120 cm. Right: Population firing rates of 1024 grid cells as a function of the movement direction. Black and light gray lines represent the results from the numerical simulations of **Equation 8** and the analytical derivation in **Equation 32**, respectively. Grid phase offsets cells are strongly clustered at $(0, 0)$ with $\kappa_s = 10$ (left insets). (**A**) Left: Subsegment of a star-like walk. Right: Population firing rate as a function of the movement direction for star-like runs originating at phase offset $(0, 0)$ with mean firing rate $\tilde{A}_0 = 1362.4$ spk/s (spikes/second) and path hexasymmetry $|\tilde{T}_6| < 10^{-10}$. (**B**) Left: Subsegment of a piecewise linear walk. Right: Population firing rate as a function of the movement direction for a piecewise linear trajectory. $\tilde{A}_0 = 1244.3$ spk/s, $|\tilde{T}_6| < 10^{-10}$. (**C**) Left: Subsegment of a random walk. Right: Population firing rate as a function of movement direction for a random walk. $\tilde{A}_0 = 1305.4$ spk/s, $|\tilde{T}_6| = 6.7 \cdot 10^{-3}$. (**D**) Left: Hexasymmetry as a function of the subject's starting location (relative to grid phase offset). Right: Movement direction associated with the highest sum grid-cell activity, i.e., the phase of peaks in (**A**, right), has a bimodal distribution (0 or 30°). (**E**) Same as (**D**) but for piecewise linear walks. (**F**) Same as (**D**) but for random walks. (**G**) Example of a two-dimensional slice of a three-dimensional

*Figure 4 continued on next page*

*Figure 4 continued*

(3D) random-field simulation with a spatial resolution of 15 µm. The simulated volume of $3 \times 3 \times 3$ mm³ represents the approximate spatial extent of a voxel in functional magnetic resonance imaging (fMRI) experiments (for details, see Methods). (**H**) The pairwise phase distance in the rhombus is shown as a function of the pairwise anatomical distance for all pairs of simulated cells from (**G**). Since no spatial correlation structure is induced, the pairwise phase distance between grid cells remains constant when varying the pairwise anatomical distance between them. (**I**) The resulting phase clustering for $200^3$ simulated grid cells in a $3 \times 3 \times 3$ mm³ voxel. Brighter colors indicate a higher prevalence of a particular grid phase offset. The distribution of grid phases appears to be homogeneous with a clustering concentration parameter of $\kappa_s = 0.001$. (**J–M**) Same as (**G–I**) but for a convolution of the 3D random field with the 3D Gaussian correlation kernel of width 30 µm shown in (**K**). Grid cells located next to each other in anatomical space ($\lesssim 30\,\mu m$) exhibit similar grid phase offsets (**L**). No clear clustering is visible, and the fitted clustering concentration parameter was $\kappa_s = 0.014$ (**M**). (**N–Q**) Same as (**J–M**) but for the grid-like correlation kernel with projections shown in (**O**), which adds some longer-range spatial autocorrelation to the grid phase offsets of different grid cells. The red bars in (**O**) measures a distance of 300 µm, which corresponds to the separation between two peaks of the correlation kernel lying on the same plane. The pairwise phase distance (**P**) exhibits a dip around 300 µm. Note in (**Q**) that the prevalence of particular grid phase offsets is more biased than in (**M**), with a fitted clustering concentration parameter of $\kappa_s = 0.052$. (**R**) Dependence of the clustering concentration parameter $\kappa_s$ on the number $N$ of grid cells in a voxel. A random distribution of grid cells in anatomical space was obtained by subsampling from the $200^3$ grid cells simulated in (**G–Q**) over 300 realizations. We found $\kappa_s \approx 0.05$ for $10^3$ grid cells in a voxel, and that $\kappa_s$ decreases monotonically as $N$ is increased. Convolution of grid phase offsets with a correlation kernel (as in **J–Q**) leads to saturation of $\kappa_s$ for large $N$. Note that the range of values of $\kappa_s$ here is three orders of magnitude smaller than the strong clustering considered in (**A–C**). The dashed-dotted line depicts the line $1/\sqrt{N}$ for comparison. The vertical dotted line at $N = 20^3$ corresponds to the empirically estimated count of grid cells in a $3 \times 3 \times 3$ mm³ fMRI voxel. The secondary lower horizontal axis shows the average distance between regularly distributed grid cells in anatomical space. (**S**) The distribution of the clustering concentration parameter $\kappa_s$ when using either no kernel, a Gaussian kernel, or a grid-like kernel for $20^3$ grid cells subsampled from the $200^3$ grid cells simulated in (**G–Q**) over 300 realizations. The grid-like kernel results in a larger maximum value of the clustering concentration parameter, and a large number of realizations results in relatively low clustering. \*\*\*, p<0.001. (**T**) The dependence of the hexasymmetry $|\tilde{A}_6|$ on the number of angles sampled when unwrapping the star for the piecewise linear walk, averaged over 20 trajectories for each data point. Each additional angle sampled adds 300 cm to the total length of the path. A line proportional to $c/\sqrt{M}$ is plotted for comparison, where $c$ is an offset parameter ($c = 4000$ spk/s) chosen such that the slope of $|\tilde{A}_6|$ can be compared to the slope of $1/\sqrt{M}$, which is proportional to the path hexasymmetry $|\tilde{T}_6|$ (*Equation 67*). The close fit between the solid and dashed lines indicates that the neural hexasymmetry $|\tilde{A}_6|$ is highly correlated with the path hexasymmetry $|\tilde{T}_6|$. The vertical dotted line at 360 sampled angles corresponds to the number of angles sampled in the star-like run and the piecewise linear run for all main figures. The value of the clustering concentration parameter used was $\kappa_s = 10$.

The online version of this article includes the following figure supplement(s) for figure 4:

**Figure supplement 1.** Comparison of hexasymmetry resulting from the standard grid cell model ('standard') and the structure-function mapping hypothesis ('Gu', identical to *Figure 5*).

the grid-phase offsets (*Figure 4N–Q*; *Gu et al., 2018*), which led to a clustering concentration parameter $\kappa_s \approx 0.052$ (*Figure 4Q*). Accordingly, we found that these more 'realistic' clustering concentration parameters resulted in substantially reduced hexasymmetries for the structure-function mapping hypothesis (described in the next section and in *Figure 5*). Furthermore, the neuronal hexasymmetries were much lower than the ones for the 'realistic' conjunctive grid by HD cell hypothesis and for the 'realistic' repetition suppression hypothesis.

Our above-mentioned results for the structure-function mapping hypothesis indicate that the navigation pattern has a strong influence on the neuronal hexasymmetry $H$ (*Figure 4A–C*). We thus wondered how much the hexasymmetries of the associated navigation paths contributed to the neuronal hexasymmetries. We first considered random walks and found that the path hexasymmetry was proportional to $1/\sqrt{M}$ for a large number $M$ of steps (*Figure 1—figure supplement 4*). For the random walk in *Figure 4C* with tortuosity $\sigma_\theta = 0.5$ rad/s$^{1/2}$, we derived from *Figure 1—figure supplement 4* that the expected path hexasymmetry for 9000 s simulation time ($M = 0.9 \cdot 10^6$ steps for $\Delta t = 0.01$ s) is about 0.007, which results in a contribution to the neural hexasymmetry of $H \approx 7$ spk/s for a population rate of about 1024 spk/s. This number is close to the obtained neural hexasymmetry ($H = 8.0$ spk/s) in *Figure 4C*. We thus believe that for random walks the hexasymmetry $H$ obtained in the structure-function mapping case is mainly determined by the path hexasymmetry.

We then turned to piecewise linear walks and examined neural hexasymmetries $H$ across a broad range of total trajectory distances. Surprisingly, we observed that $H$ also decreased as $1/\sqrt{M}$ (*Figure 4T*) even though the path hexasymmetry of piecewise linear walks is basically zero by construction. This indicates that the apparent neural hexasymmetry $H$ of sum grid-cell activity for piecewise linear walks (e.g. in *Figure 4B*) is driven by random subsamples of all path segments—specifically those path segments crossing through grid fields. These subsamples of path segments necessarily exhibit higher path hexasymmetries than the full set of path segments that has zero path hexasymmetry. We thus

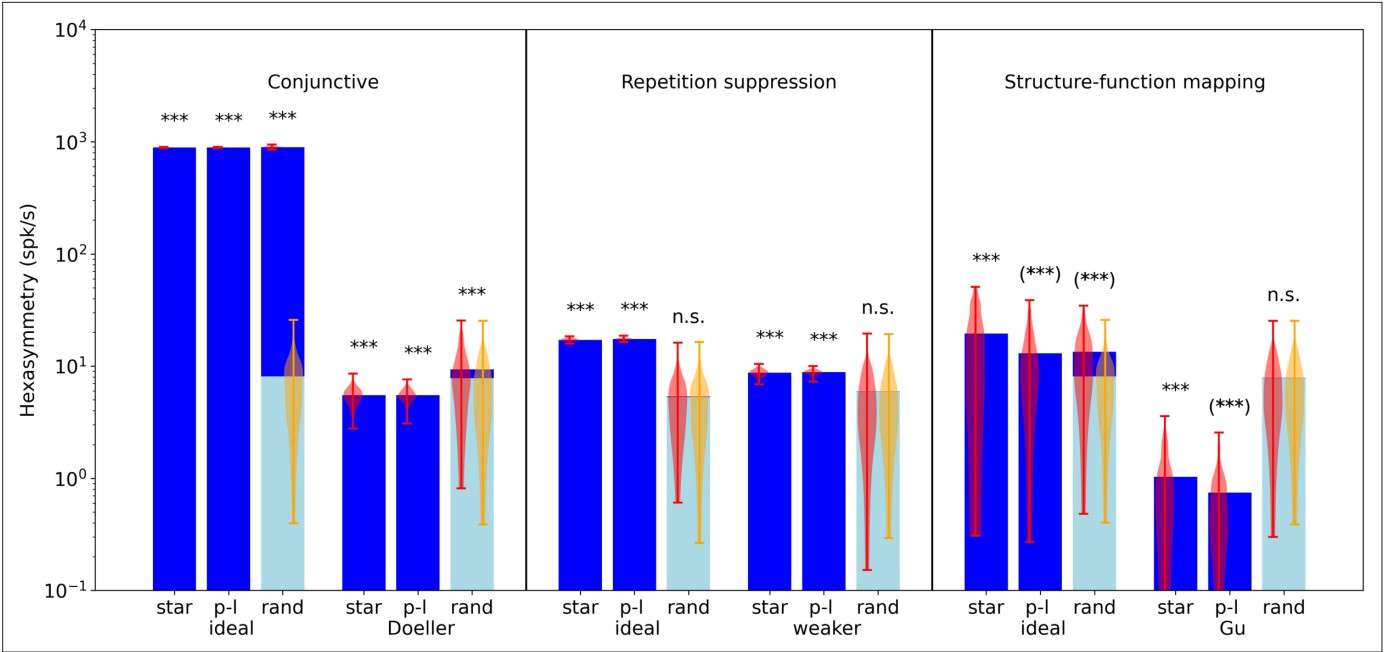

**Figure 5.** Comparison of hexasymmetry resulting from the three hypotheses. Each of the three hypotheses is implemented for three different types of navigation trajectories: star-like walks ('star'), piecewise linear walks ('p-l'), and random walks with small step size ('rand'). For each setting, we show the average neural hexasymmetry $|\tilde{A}_6|$ for 1024 cells (dark blue bars) and the average contribution of the trajectory $|\tilde{T}_6| \cdot \tilde{A}_0$ (light blue bars) where $\tilde{A}_0$ is the (variable) average population activity and $|\tilde{T}_6|$ is the hexasymmetry of the trajectory; see Methods for definitions of symbols. The violin plots depict the distributions for firing-rate hexasymmetry (red) and path hexasymmetry (orange). For each hypothesis, we calculate the hexasymmetry for 'ideal' parameters (conjunctive: $p_c = 1$, $\kappa_c = 50\ \mathrm{rad}^{-2}$, tuning width $1/\sqrt{\kappa_c} = 8.1^{\circ}$, $\sigma_c = 0$; repetition suppression: $\tau_r = 3$ s, $w_r = 1$; clustering: $\kappa_s = 10$) as well as more realistic parameters (conjunctive 'Doeller': $p_c = 0.33$, $\kappa_c = 4\ \mathrm{rad}^{-2}$, tuning width $1/\sqrt{\kappa_c} = 29^{\circ}$, $\sigma_c = 3^{\circ}$ motivated by *Doeller et al., 2010*; *Boccara et al., 2010*; *Sargolini et al., 2006*; repetition suppression 'weaker': $\tau_r = 1.5$ s, $w_r = 0.5$; clustering 'Gu': $\kappa_s = 0.1$ motivated by *Gu et al., 2018*). For clustering, we also simulated grid phase offsets randomly sampled from a uniform distribution with $\kappa_s = 0$ and found that neural hexasymmetries were reduced (compared to 'Gu') by $\sim 30\%$ (*Figure 4—figure supplement 1*). The parameters for the random-walk scenario are $T = 9000$ s and $\Delta t = 0.01$ s; see *Table 1* for a summary of parameters. Each hypothesis condition was simulated for 300 realizations. \*\*\*, p<0.001; n.s., not significant; (\*\*\*), a seemingly significant result (p<0.001) that we think is spurious (see Results section). For pairwise comparisons of the hexasymmetry values from different trajectory types for each set of parameters, see *Figure 5—figure supplement 1*. For a quantification of path hexasymmetry as a function of the percentage of subsampled path segments, see *Figure 5—figure supplement 2*. For a comparison between our method and previously used methods for evaluating hexasymmetry, see *Figure 5—figure supplement 4*.

The online version of this article includes the following figure supplement(s) for figure 5:

**Figure supplement 1.** Pairwise comparisons of the hexasymmetry values from different trajectory types for each set of parameters.

**Figure supplement 2.** Hexasymmetry resulting from piecewise linear walks ('p-l') and random walks ('rand') as a function of the percentage of subsampled path segments for the structure-function mapping hypothesis.

**Figure supplement 3.** Comparison of the hexasymmetry resulting from the three hypotheses when using trajectories from rats and humans.

**Figure supplement 4.** Comparison of hexasymmetry resulting from the three hypotheses when using different methods to calculate the hexasymmetry.

conclude that, for piecewise linear walks, hexasymmetry values were driven by a subsampling of the movement directions due to the sparsity of the grid-field locations.

Taken together, the structure-function mapping hypothesis with very strong ('ideal') clustering produces neural hexasymmetry values that are larger than expected from path hexasymmetries only with respect to star-like walks, including a strong dependence on the subject's starting location (*Figure 4D*, left). This range of hexasymmetry values is comparable to those of the repetition suppression hypothesis with 'ideal' parameters (*Figure 3*), but values are at least an order of magnitude smaller than in the 'ideal' conjunctive grid by HD cell case (*Figure 2*). When using realistic clustering concentration parameters, neural hexasymmetry values are further reduced as compared to simulations with ideal clustering (*Figure 5*).

## Overall evaluation of the three hypotheses

To provide a systematic evaluation of the three hypotheses, we computed 300 realizations of each hypothesis (using the simulated activity of 1024 cells), separately for each type of navigation and for both ideal and more realistic parameter settings (see *Table 1* for a summary of parameter values and section 'Parameter estimation' for their justification). This resulted in 18 different hypothesis conditions (*Figure 5*; see also *Figure 5—figure supplement 1* for pairwise comparisons). For each hypothesis condition, we assessed its statistical significance by performing nonparametric Mann-Whitney U tests between the neural hexasymmetries ($H := |\tilde{A}_6|$; see also *Equation 12*) and the product $|\tilde{T}_6| \cdot \tilde{A}_0$ with the multipliers path hexasymmetry $|\tilde{T}_6|$ and average population activity $\tilde{A}_0$.

For the conjunctive hypothesis, we found that all three types of navigation led to significant neural hexasymmetries. This was the case for both ideal parameters (Mann-Whitney U tests, all $U=0$, all p<0.001) and for more realistic parameters (Mann-Whitney U tests, all $U=0$, all p<0.001). For star-like walks and piecewise linear walks, the path hexasymmetries multiplied with the average population activities were near zero because we designed these navigation paths to equally cover all different movement directions. For random walks, the path hexasymmetries multiplied with the average population activities were at values of about 7 spk/s and showed larger variability because the directions of the random walks were not predefined and could thus vary with regard to their hexasymmetries. If the conjunctive hypothesis was true, fMRI studies should thus see a hexadirectional modulation of entorhinal fMRI activity for all three types of the subjects' navigation paths.

For the repetition suppression hypothesis, we found that star-like walks and piecewise linear walks resulted in significant neural hexasymmetries for both ideal and weaker parameters (Mann-Whitney U tests, all $U=0$, all p<0.001), whereas random walks did not (Mann-Whitney U tests, both $U>1753$, both p>0.405). For random walks, the path hexasymmetries multiplied with the average population activities were lower as compared to the other two hypotheses because the repetition suppression necessarily leads to lower average population activities. If the repetition suppression hypothesis was true, fMRI studies should thus observe significant neural hexasymmetries only for star-like walks and piecewise linear walks with long enough segments, whereas random walks (with large enough tortuosities, *Figure 1—figure supplement 1*) should not lead to significant neural hexasymmetries.

Regarding the structure-function mapping hypothesis, the statistical tests showed that most of the hypothesis conditions resulted in significant neural hexasymmetries as compared to the path hexasymmetries multiplied with the average population activities. This was the case for both ideal parameters (Mann-Whitney U tests, all $U<736$, all p<0.001) and more realistic parameters (Mann-Whitney U tests for star-like walks and piecewise linear walks, all $U=0$, all p<0.001). Only the hypothesis condition with random walks and more realistic parameters exhibited no significant neural hexasymmetries (Mann-Whitney U test, $U=1786$, p=0.472). However, these results regarding the structure-function mapping hypothesis should be treated with great caution. First, the neural hexasymmetries for star-like walks heavily depend on the starting location of the subject relative to the grid fields, and different starting locations lead to different apparent grid orientations (*Figure 4D*). Second, the significant results for the navigation conditions with piecewise linear walks and random walks actually result from an inhomogeneous sampling of movement directions through the grid fields and therefore do not reflect true neural hexasymmetries (*Figure 4T*; see also *Figure 5—figure supplement 2* for path hexasymmetry as a function of the number of subsampled path segments). Only the structure-function mapping hypothesis is susceptible to this effect because the grid fields are clustered at similar spatial locations (whereas the grid fields are homogeneously distributed in the case of the conjunctive hypothesis and the repetition suppression hypothesis). In simulations with infinitely long paths, the neural hexasymmetries (for the navigation types of piecewise linear walks and random walks) would not be significantly higher than the path hexasymmetries multiplied with the average population activities. In empirical studies, this effect can be detected by correlating the subject-specific path distances with the subject-specific neural hexasymmetries: if there is a generally negative relationship, this will hint at the fact that the neural hexasymmetries are basically due to relevant path hexasymmetries of path segments crossing the grid fields. In essence, we therefore believe that the structure-function mapping hypothesis leads to true neural hexasymmetry only in the case of star-like walks.

## Influence of other factors

Our simulations above were performed in an infinite spatial environment, which is different from empirical studies in which subjects navigate finite environments. We were thus curious whether the size and shape of finite environments could affect the strength of hexadirectional modulations of population grid-cell activity.

Our simulations showed that for both circular and square environments, hexasymmetry strengths did not considerably depend on the size of the environment when the subject performed random walks (*Figure 1—figure supplement 2*). Similarly, rotating the navigation trajectories relative to the grid patterns did not affect the hexasymmetry strengths (*Figure 1—figure supplement 3*). These results thus suggest that experiments in animals and humans can use various types and sizes of the environments to investigate hexadirectional modulations of sum grid-cell activity.

## Discussion

We performed numerical simulations and analytical estimations to examine how the activity of grid cells could potentially lead to a neural population signal in the EC. Such a neural population signal has been observed in multiple fMRI studies including *Doeller et al., 2010*; *Kunz et al., 2015*; *Constantinescu et al., 2016*; *Horner et al., 2016*; *Bellmund et al., 2016*; *Stangl et al., 2018*; *Nau et al., 2018*; *Julian et al., 2018*; *Bierbrauer et al., 2020*; *Julian and Doeller, 2021*; *Bongioanni et al., 2021*; *Moon et al., 2022*, and in multiple iEEG/MEG studies including *Maidenbaum et al., 2018*; *Chen et al., 2018*; *Staudigl et al., 2018*; *Chen et al., 2021*; *Wang and Wang, 2021*; *Convertino et al., 2023*. The recorded population signals showed a hexadirectional modulation of the entorhinal fMRI/iEEG signal as a function of the subject's movement direction through its spatial environment. We note though that some studies did not find evidence for neural hexasymmetry. For example, a surface EEG study with participants 'navigating' through an abstract vowel space did not observe hexasymmetry in the EEG signal as a function of the participants' movement direction through vowel space (*Kaya et al., 2020*). Another fMRI study did not find evidence for grid-like representations in the ventromedial prefrontal cortex while participants performed value-based decision-making (*Lee et al., 2021*). This raises the question whether the detection of macroscopic grid-like representations is limited to some recording techniques (e.g. fMRI and iEEG but not surface EEG) and to what extent they are present in different tasks.

### Potential mechanisms underlying hexadirectional population signals in the EC

We examined three hypotheses that have been previously suggested as potential mechanisms underlying the emergence of the hexadirectional population signal in the EC (*Doeller et al., 2010*; *Kunz et al., 2019*) and found that all three hypotheses can—in principle and in ideal situations—lead to a hexadirectional modulation of EC population activity. However, none of the three hypotheses described here may be true and another mechanism may explain macroscopic grid-like representations. This includes the possibility that neural hexasymmetry is completely unrelated to grid-cell activity, previously summarized as the 'independence hypothesis' (*Kunz et al., 2019*). For example, a population of HD cells whose preferred head directions occur at offsets of 60° from each other could result in neural hexasymmetry in the absence of grid cells. The conjunctive grid by HD cell hypothesis thus also works without grid cells, which may explain why grid-like representations have been observed (using fMRI) in regions outside the EC (*Doeller et al., 2010*; *Constantinescu et al., 2016*), where rodent studies have not yet identified grid cells except for some evidence of grid cells in the primary somatosensory cortex of foraging rats (*Long and Zhang, 2021*). In that case, however, another mechanism would be needed that could explain why the preferred head directions of different HD cells occur at multiples of 60°. Attractor-network structures may be involved in such a mechanism, but this remains speculative at the current stage.

### Navigation paths have a major influence on the hexadirectional population signal

A major observation of this study is that the way how subjects navigate through the environment has a major influence on whether a hexadirectional population signal can be observed. We distinguished

three major types of navigation: navigation with random-walk trajectories in which straight paths are quite short, which resembles the navigation pattern in rodents; navigation with piecewise linear trajectories in which the subject navigates along straight paths combined with random sharp turns between the straight segments; and navigation with star-like trajectories in which the subject starts each path from a fixed center location and navigates along a straight path with a predetermined allocentric direction and distance. Critically, we found that the conjunctive hypothesis leads to a hexadirectional population signal irrespective of the specific type of navigation; that the repetition suppression hypothesis leads to hexadirectional population signals only in the case of star-like trajectories and piecewise linear trajectories (but not for random trajectories); and that for the structure-function mapping hypothesis 'true' hexadirectional population signals can only be observed for star-like trajectories.

The finding that the type of navigation paths influences whether a hypothesis can lead to hexadirectional population signals of the EC is informative to future fMRI/iEEG studies, which could empirically evaluate which of the three hypotheses is most likely to be true: By asking or requiring the subjects to navigate in different ways through the task environments (in a star-like fashion, in a piecewise linear fashion, and in a random fashion), these future fMRI/iEEG studies could demonstrate whether hexadirectional population signals are present in all three navigation conditions (speaking in favor of the conjunctive hypothesis); whether they are present only during star-like or piecewise linear trajectories (in favor of the repetition suppression hypothesis); or whether they are mainly visible for star-like trajectories and exhibit a systematic decrease with increasing total trajectory length for piecewise linear and random walks (in this case speaking in favor of the structure-function mapping hypothesis). We note that in our simulations, each linear path segment of a piecewise linear walk has a length that is 10 times larger than the grid scale. While humans performing navigation tasks exhibit straighter trajectories than those of rats (*Doeller et al., 2010*; *Kunz et al., 2015*; *Horner et al., 2016*), they are still more similar in scale to random walks than piecewise linear walks. Thus, when using empirical navigation paths from previous studies, neural hexasymmetries were significant only for the conjunctive hypothesis (*Figure 5—figure supplement 3*). In contrast to this major effect of navigation type on the presence of hexadirectional signals, the size and shape of the environment did not influence the strength of the hexadirectional signals in a relevant manner (*Figure 1—figure supplement 2* and *Figure 1—figure supplement 3*).

The structure-function mapping hypothesis predicts that in fMRI studies involving star-like walks, there may be up to 30° shifts in the orientation of hexadirectional modulation between neighboring voxels (*Figure 4D*). This is in contrast with the other two hypotheses, where the preferred grid orientations are similar between voxels, and may differ only slightly when recording distinct grid modules with different grid orientations (*Stensola et al., 2012*).

## A note on our choice of the values of model parameters

Another insight of this study is that the exact biological properties of grid cells play a major role regarding the question whether hexadirectional population signals can be observed. For example, the conjunctive hypothesis cannot lead to hexadirectional population signals if the tuning width of the conjunctive grid by HD cells is too broad (*Figure 2E*). Here, the percentage of strongly tuned conjunctive cells also plays a relevant role. Empirical studies in rodents found a wide range of tuning widths among grid cells ranging from broad to narrow (*Doeller et al., 2010*; *Sargolini et al., 2006*). The percentage of conjunctive cells in the EC with a sufficiently narrow tuning may thus be low. Such distributions (with a proportionally small amount of narrowly tuned conjunctive cells) lead to low values in the absolute hexasymmetry. The neural hexasymmetry in this case would be driven by the subset of cells with sufficiently narrow tuning widths. If this causes the neural hexasymmetry to drop below noise levels, the statistical evaluation of this hypothesis would change.

We also found that hexadirectional population signals emerge only if the preferred directions of the conjunctive cells are aligned precisely enough with the grid axes of the grid cells (*Figure 2E*). Furthermore, no hexadirectional population activity will emerge if the preferred head directions of conjunctive grid by HD cells exhibit other types of rotational symmetry such as 10-fold rotational symmetry reported in recordings from rats (*Keinath, 2016*). Additionally, while we assumed that all conjunctive grid cells maintain the same preferred head direction between different firing fields, conjunctive grid cells have also been shown to exhibit different preferred head directions in different firing fields (*Gerlei et al., 2020*). This could lead to hexadirectional modulation if the different preferred head

directions are offset by 60° from each other, but will not give rise to hexadirectional modulation if the preferred head directions are randomly distributed. To the best of our knowledge, the distribution of preferred head directions was not quantified by *Gerlei et al., 2020*, thus this remains an open question. Together, it currently seems possible that grid cells and conjunctive grid by HD cells meet the necessary biological properties for the conjunctive hypothesis (*Doeller et al., 2010*), but further studies on the characteristics of conjunctive grid by HD cells (also in humans) will be useful because they may find tuning widths which are too wide or tuning curves which are not aligned with the grid axes.

The exact biological properties of grid cells also play a major role for the structure-function mapping hypothesis, which heavily relies on the property of neighboring grid cells to share a similar grid phase offset (i.e. a high spatial autocorrelation of grid phases) and whether there might be longer-range spatial autocorrelations between the grid phases (*Heys et al., 2014*; *Gu et al., 2018*). With the evidence at hand, it seems unlikely to us that the grid phases are clustered strongly enough to facilitate a hexadirectional population signal. Specifically, we found that the clustering concentration parameters of grid phase offsets gleaned from existing empirical studies (*Gu et al., 2018*) produced the smallest neural hexasymmetry when compared to the other conditions we considered, while the neural hexasymmetries obtained from simulations randomly sampling grid phase offsets from a uniform distribution on the unit rhombus (that one might refer to as the 'standard grid cell model') were only slightly smaller.

Regarding realistic parameters for the repetition suppression hypothesis, we are currently not aware of detailed measurements of the relevant grid-cell properties (i.e. the adaptation time constant $\tau_r$ and the adaptation weight $w_r$) so that it remains unclear to us to what extent the repetition suppression hypothesis is biologically plausible. Future studies may thus quantify the relevant properties of (human) grid cells in greater depth to help clarify which hypothesis regarding the emergence of hexadirectional population signals may most likely be true.

## Influence of grid scale on hexasymmetry

In this study, we assumed that all grid cells have the same grid spacing of 30 cm. For rats, this is at the lower end of grid spacing values, corresponding to the dorsal region of the medial EC (*Stensola et al., 2012*). How would the three hypotheses perform for larger values of grid spacing? The conjunctive hypothesis would remain unchanged as it does not depend on grid spacing. In contrast, the repetition suppression hypothesis depends on the grid scale. Here, the strongest hexasymmetry is achieved when the adaptation time constant is similar to the average time that it takes the subject to move between neighboring grid fields (*Figure 3B*). Altering the grid scale changes this traversal time between grid fields and thus hexasymmetry. In *Figure 3*, we chose the 'ideal' adaptation time constant to maximize hexasymmetry. A change in grid scale would thus reduce hexasymmetry for this particular value of the adaptation time constant. As there are different grid-cell modules in the EC that exhibit different grid spacings (*Stensola et al., 2012*), repetition suppression effects may vary across modules (if the adaptation time constant is not tuned in accordance with the grid scale). Regarding the structure-function mapping hypothesis and star-like walks, different grid spacings can have a major effect on neural hexasymmetry if the path segments are short compared to the grid scale, similar to effects of the starting location (*Figure 4D*); for longer path segments, altering the grid scale should have a smaller effect on neural hexasymmetry. For the structure-function mapping hypothesis, random walks and piecewise-linear walks do not lead to real neural hexasymmetries, and different grid spacings are thus irrelevant. Overall, different grid spacings are thus most important in the context of the repetition suppression hypothesis.

Another factor that might complicate the picture is an interaction of different grid scales: rodent grid cells have been found to be organized into four to five discrete modules with different grid scales (*Stensola et al., 2012*) and this may also be true in humans. Together with the anatomical dorsoventral length of the human EC ($\approx$2 cm, *Behuet et al., 2021*), this means that a module might cover roughly 4–5 mm. A typical voxel of $3 \times 3 \times 3$ mm$^3$ would thus either contain one or two grid modules. The case of a voxel containing a single module was already discussed. If a voxel contained two modules, there would be two subpopulations of grid cells with different grid spacings: For the structure-function mapping hypothesis, significant neural hexasymmetry can be achieved only for star-like trajectories. In this scenario, the hexasymmetry depends on the starting position of the trajectories, i.e., the center

position of the star (*Figure 4D*). Two subpopulations of grid cells from different modules essentially act like two individual grid cells in the context of the structure-function mapping hypothesis. Thus, they can either add their hexasymmetry values (e.g. when the star center is in the middle of a grid field for both subpopulations) or suppress each other (when the star center is in a regime of 0° grid phase for one subpopulation and in a regime of ±30° grid phase for the other). For the repetition suppression hypothesis, two populations of grid cells with the same adaptation time constant but different grid spacing would result in two different values of hexasymmetry, depending on the interaction between grid-field traversal time and adaptation time constant. The joint hexasymmetry value would then depend on the two subpopulation sizes. We note that the adaptation time constant might be tuned to match the grid spacing. This would result in high hexasymmetry values for both subpopulations, and thus also for the full population.

## Sources of noise

The neural hexasymmetry as defined by *Equation 12* also contains contributions from the path hexasymmetry of the underlying trajectory. Star-like walks and piecewise linear walks have a path hexasymmetry of zero by construction. Therefore, these trajectory types do not contribute to the neural hexasymmetry. For random walks, the path hexasymmetry depends on the number of time steps $M$ and the tortuosity parameter $\sigma_\theta$ (*Figure 1—figure supplement 4*). To compare the path hexasymmetry to the neural hexasymmetry, the path hexasymmetry needs to be multiplied by the time-averaged population activity. Since the path hexasymmetry ranges from 0 to 1 (*Equation 14*), its contribution to the neural hexasymmetry ranges from 0 (for a uniform sampling of movement directions) to the time-averaged population activity. For $M$=900,000 time steps and a tortuosity of $\sigma_\theta = 0.5$ rad/s$^{1/2}$, random-walk trajectories contribute ~0.8% of the time-averaged population activity to the neural hexasymmetry (*Figure 1—figure supplement 4*). In our simulations with an average population activity in the range of $10^3$ spk/s, the 0.8% noise floor leads to a path hexasymmetry of about 8 spk/s (*Figure 5*).

Another factor that contributes noise is the finite sampling of grid phase offsets. For all our simulations of the conjunctive grid by HD cell hypothesis and the repetition suppression hypothesis (and for control simulations of the structure-function mapping hypothesis), we sample a finite number of grid phase offsets from a two-dimensional von Mises distribution with a clustering concentration parameter $\kappa_s = 0$ (equivalent to a uniform distribution on the unit rhombus). In this scenario, random fluctuations give rise to realizations with a non-zero sample mean clustering concentration parameter $\hat{\kappa}_s$, which led to a corresponding mean neural hexasymmetry of 0.7 spk/s (labeled 'standard' and 'star' in *Figure 4—figure supplement 1*), even in the absence of conjunctive tuning and repetition suppression. For 1024 grid cells, this contribution is 10 times smaller than the neural hexasymmetry resulting from the 'realistic' parameters for both the conjunctive grid by HD cell hypothesis and the repetition suppression hypothesis (*Figure 5*). In the case of the structure-function mapping hypothesis, these fluctuations would not be considered as 'noise' as they directly contribute to the hexasymmetry from the clustering of grid phase offsets.

## On the relation of firing rates and fMRI/iEEG signals

A topic that this study did not investigate is the question of how the sum signal of single neurons translates into fMRI and iEEG signals. In neocortical regions such as the auditory cortex, a clear linear relationship between single-neuron activity and fMRI activity has been observed (*Mukamel et al., 2005*), but it remains elusive whether this linear relationship also applies to the EC in general and to entorhinal grid cells in particular. In the neighboring hippocampus, for example, the relationship between single-neuron activity and the fMRI signal is highly complex (*Ekstrom, 2010*; *Kunz et al., 2019*). Future studies are needed to detail the relationship between single-neuron firing and fMRI/iEEG signals in the EC. This would allow us to clarify whether a hexadirectional modulation of sum grid-cell activity directly results in a hexadirectional modulation of fMRI/iEEG activity or whether currently unknown factors modulate the expression of hexadirectional fMRI/iEEG signals.

## Conclusion

Using numerical simulations and analytical derivations we showed that a hexadirectional neural population signal can emerge from the activity of grid cells given the ideal conditions of three different

hypotheses. Whether a given hypothesis leads to a hexadirectional population signal is significantly influenced by the subjects' type of navigation through the spatial environment and by the exact biological properties of human grid cells.

## Methods
### Trajectory modeling
To describe grid-cell activity as a function of time $t$ during which a subject (animal or human) is exploring an environment, we model three distinct trajectory types. We first describe trajectory types in environments without bounds, which are quasi-infinite, and then add rules that account for boundaries.

#### Environments without boundaries
The first trajectory type is a random walk $X_t = [x_t, y_t]$, which is defined by

$$\frac{\mathrm{d}X_t}{\mathrm{d}t} = v \left[ \cos(\theta_t), \sin(\theta_t) \right] \tag{1}$$

with $\theta_t = \sigma_\theta \cdot W_t$, where $\sigma_\theta$ controls the tortuosity of the trajectory and $W_t$ is a standard Wiener process. In a numerical simulation with a time step $\Delta t$, the angle is updated in each time step by $\theta_{t+\Delta t} = \theta_t + \sigma_\theta \cdot \Delta W$, where $\Delta W$ is a normally distributed random variable with variance $\Delta t$. The variable $v$ depicts the (constant) speed.

The second type of navigation is a star-like walk, where the subject moves radially outward from a predefined origin in space at a certain angle $\theta$ on a straight line to a maximum distance $r_{max}$ at a constant speed $v$. In simulations, this movement is repeated (with the same predefined origin) for $N_\theta$ angles that are equally spaced on the interval $[0, 2\pi)$. Within each individual radial path, the subject does not turn around and move back to the origin, i.e., the entire trajectory of $N_\theta$ radial paths is not continuous.

Finally, we introduce a piecewise linear walk, which is constructed by placing all the radial paths of the star-like walk end-to-end such that they form one single continuous trajectory of length $N_\theta r_{\max}$. The trajectory thus consists of successive straight runs for the simulated subject, which can be interpreted as a random walk with a time step $\Delta t = r_{\max}/v$ and directions that are sampled uniformly without replacement from a predetermined set of angles. In comparison to the random walk and the star-like walk, this procedure presumably reflects the situation in human virtual-reality setups most closely, as participants often move along straight trajectories with intermittent turns (*Doeller et al., 2010*; *Kunz et al., 2015*; *Horner et al., 2016*).

#### Environments with boundaries
Most virtual-reality studies in humans use finite instead of infinite spatial environments to examine grid-like representations. We wondered whether the size and the shape of these finite environments might modulate the strength of macroscopic grid-like representations obtained through one or more of the three hypotheses. Hence, we performed simulations not only in infinite but also in finite environments with a given size (between one and six times the grid spacing) and shape (circle and square).

For random-walk trajectories, we enforce that the navigating subject stays within the circular or square environment by performing an 'out-of-bounds check' at each time point. This means that, after every time step $\Delta t$, we measure the distance that the subject has moved outside of the boundary. This is done differently in square and circular environments, both of which are centered at the origin ($x = 0, y = 0$). In the square environment, we define the variables $\Delta x$ and $\Delta y$ as the distance the subject has moved out of bounds in the $x$ and $y$ coordinates, respectively. Let $L$ be the half of the length of the square's sides. $\Delta x$ and $\Delta y$ are then defined as $\Delta x = \max \left[ |x| - L, 0 \right]$ and $\Delta y = \max \left[ |y| - L, 0 \right]$. For circular environments, let $R$ be the radius of the circle, and let $\vec{r} = (x, y)$ be the position vector of the subject. We then introduce the measure $\Delta r := \max \left[ \|\vec{r}\| - R, 0 \right]$, such that $\Delta r$ is non-zero only when the subject has moved outside of the circular boundary. If at any time point either $\Delta x$, $\Delta y$, or $\Delta r$ are non-zero, the out-of-bounds check fails. In this case, we reject the movement in this time step and keep resampling a new angle $\theta_t$ (*Kropff and Treves, 2008*; *Si et al., 2012*) until the check succeeds, meaning that the subject has made a move that is within the boundaries of the environment. If $\Delta x$, $\Delta y$, or $\Delta r$ remain non-zero for 50 consecutive samples of $\theta_t$, we temporarily increase the tortuosity $\sigma_\theta$ by

a factor 1.1. Without this increase in tortuosity, the subject tends to get stuck when approaching the boundary at angles close to the perpendicular or at the corners of the square boundary. The tortuosity is reset to its initial value once a valid move is made. We visually checked the random-walk trajectories, which show some oversampling along the boundaries, and found that they were comparable to the navigation trajectories in rodent studies (e.g. *Hafting et al., 2005*).

## Implementation of grid cell activity

Following *Burgess et al., 2007*, the activity profile $G_i$ of grid cell $i$ (for $i = 1, ..., N$ in a population of $N$ grid cells) is modeled as the product of three cosine waves rotated by 60° (= $\pi/3$) from each other:

$$G_i(x, y) = \frac{A_{\max}}{8} \prod_{k=0}^{2} \left( 1 + \cos \left[ \frac{4\pi}{\sqrt{3}s_i} \sin \left( \frac{\pi}{3}k + \gamma_i \right) (x - s_i x_{\text{off},i}) + \frac{4\pi}{\sqrt{3}s_i} \cos \left( \frac{\pi}{3}k + \gamma_i \right) (y - s_i y_{\text{off},i}) \right] \right) \quad (2)$$

where $A_{\max}$ is the grid cell's maximal firing rate, $s_i$ depicts the cell's grid spatial scale ('grid spacing'), $x_{\text{off},i}$ and $y_{\text{off},i}$ are the phase offsets of the grid ('grid phase') in the two spatial dimensions (called $x$ and $y$ here), and $\gamma_i$ is the orientation of the grid ('grid orientation'); see *Table 1* for numerical values of parameters and *Figure 1A* for an illustration of the three grid characteristics.

To describe the activity of many grid cells (e.g. in a voxel for an MRI scan), we sum up the firing rates of $N$ grid cells in *Equation 2*. For a given trajectory $X_t = [x_t, y_t]$, the macroscopic activity as a function of time $t$ is then simply described by the sum $\sum_{i=1}^{N} G_i(x_t, y_t)$.

## Implementation of the three hypotheses to explain macroscopic grid-like representations

Here, we summarize how the activity in a population of $N$ grid cells can be described if they also exhibit (i) HD tuning, (ii) repetition suppression (i.e. firing-rate adaptation), or (iii) grid phases that are clustered across grid cells.

In all our models, the activity $G_i$ of a grid cell is described by *Equation 2*. Different grid cells typically have different phase offsets ($x_{\text{off},i}$, $y_{\text{off},i}$) but the same grid spacing $s_i := s$, $\forall i$ and grid orientation $\gamma_i := \gamma$, $\forall i$ (*Hafting et al., 2005*; *Boccara et al., 2010*; *Stensola et al., 2012*; *Gardner et al., 2022*). We assumed that all grid cells have the same grid spacing of 30 cm.

### Conjunctive grid by HD cell hypothesis

To include HD tuning in our model, we note that a given trajectory $X_t$ has an angle $\theta_t$ at time $t$. The summed firing rate, i.e., the population activity, $A^c$ from $N$ such conjunctive cells can then be described by

$$A^c(t) = \sum_{i=1}^{N} G_i(x_t, y_t) h_i(\theta_t) \quad (3)$$

where the upper index 'c' indicates 'conjunctive' and where we incorporate conjunctive grid-HD tuning via the (scaled by factor $2\pi$) von Mises distribution

$$h(\theta) = \frac{1}{I(\kappa_c)} \exp(\kappa_c \cos(\theta - \mu_c)) \quad (4)$$

with concentration parameter $\kappa_c$ and preferred angle $\mu_c$. The symbol $I$ represents the modified Bessel function of the first kind of order 0. The parameter $\kappa_c$ describes the width of the HD tuning: if $\kappa_c \longrightarrow \infty$, the HD tuning is sharpest; the smaller $\kappa_c$, the wider the HD tuning (see *Figure 2*); for $\kappa_c = 0$, there is no HD tuning, and our scaling leads to $h(\theta) \equiv 1$. We choose the preferred angle as $\mu_c = \frac{\pi}{3}k + \eta$, where $k$ is randomly drawn from $\{0, 1, 2, 3, 4, 5\}$ and $\eta$ is randomly drawn from a normal distribution with mean 0 and standard deviation $\sigma_c$. For $\sigma_c = 0$, the directional tuning is thus centered around a multiple of 60°. The parameter $\sigma_c$ introduces jitter in the alignment of directional tuning to one of the grid axes.

We modeled the cases in which all grid cells show HD tuning ('ideal' case, fraction of conjunctive cells: $p_c = 1$) as well as a more 'realistic' case in which only a third of the cells is conjunctive ($p_c = 1/3$; *Boccara et al., 2010*; *Sargolini et al., 2006*). We note that this is an approximation, since the

proportion of conjunctive cells is highly variable across layers of the EC, with up to 90% conjunctive cells in layer V.

## Repetition suppression hypothesis

To incorporate repetition suppression in the model, we add an explicit dependence of grid-cell activity on time $t$. Specifically, we subject the firing rate $G_i$ of a grid cell to an adaptation mechanism:

$$
G_i^r(x_t, y_t, t) = \max\left[G_i(x_t, y_t) - w_r\, a(t), 0\right]
$$
$$
\tau_r \frac{\mathrm{d}a}{\mathrm{d}t} = G_i(x_t, y_t, t) - a(t),
\tag{5}
$$

where $a$ depicts the adaptation variable, and $\tau_r$ and $w_r$ are the repetition suppression time constant and the weight of the suppression, respectively. The upper index '$r$' in $G_i^r$ indicates 'repetition suppression'.

The adaptation time constant $\tau_r$ is on the order of seconds. We restrict the adaptation weight $w_r$ to the interval $[0, 1]$ (**Figure 3B**). The maximum operation 'max' in **Equation 5** ensures that the output firing rate $G_i^r$ is always positive. Together, the summed firing rate $A^r$ from $N$ such adapting cells can then be described by

$$
A^r(t) = \sum_{i=1}^{N} G_i^r(x_t, y_t, t)\,.
\tag{6}
$$

We note that the explicit dependence of the firing rate $G_i^r$ of grid cell $i$ on time $t$ needs to be considered separately for every cell for repetition suppression, which makes numerical simulations more computationally expensive. In contrast, the functions $G_i$ (in **Equation 3**) and $h_i$ (in **Equation 4**) for the conjunctive hypothesis depend also on time but only implicitly via the location $[x_t, y_t]$ and the direction $\theta_t$ of movement at time $t$—and therefore the explicit time dependence of individual cells can be disregarded, which makes numerical simulations computationally cheaper.

## Structure-function mapping hypothesis

The structure-function mapping hypothesis relies on a preferred grid phase for neighboring cells. We use two possible choices for the set of grid phases $(x_{\mathrm{off},i}, y_{\mathrm{off},i})$: they are either drawn from a uniform distribution on the unit rhombus or clustered. For clustered spatial phases, we draw $x_{\mathrm{off},i}$ and $y_{\mathrm{off},i}$ independently, each from a univariate von Mises distribution (with a defined central phase $\mu_s$ and concentration parameter $\kappa_s$). For a grid phase drawn from a uniform distribution, we note that random fluctuations lead to a certain degree of clustering of the grid-phase sample (**Figure 4—figure supplement 1**). We can describe the resulting summed activity of $N$ grid cells simply by

$$
A^s(t) = \sum_{i=1}^{N} G_i(x_t, y_t)
\tag{7}
$$

with the upper index '$s$' representing 'structure-function'.

## Quantification of hexasymmetry of neural activity and trajectories

Combining the mathematical descriptions of grid cell activity for the three hypotheses ('conjunctive', 'repetition suppression', and 'structure-function'), we can denote the resulting population activity $A$ of $N$ grid cells by

$$
A(t) = \sum_{i=1}^{N} G_i^r(x_t, y_t, t)\, h_i(\theta_t)
\tag{8}
$$

where $A_i(t) := G_i^r(x_t, y_t, t)\, h_i(\theta_t)$ is the firing rate of cell $i$. To derive from $A$ the activity as a function of movement (or heading) direction $\theta_t$, we focus on time steps of length $\Delta t$ in which the trajectory is linear. In time step $m$, i.e., for time $t$ in the time interval $[t_m, t_{m+1})$ where $t_m = m\Delta t$ and $m$ is an integer, the trajectory has the fixed angle $\theta_m$. The time-discrete mean activity $\bar{A}(t_m)$ associated to this interval is the average of $A(t)$ along the linear segment of the trajectory:

$$\bar{A}(t_m) = \frac{1}{\Delta t} \int_{t_m}^{t_{m+1}} dt \sum_{i=1}^{N} G_i^r(x_t, y_t, t)\, h_i(\theta_m) = \sum_{i=1}^{N} h_i(\theta_m) \frac{1}{\Delta t} \int_{t_m}^{t_{m+1}} dt\, G_i^r(x_t, y_t, t). \tag{9}$$

The integral in *Equation 9* is either calculated analytically, as derived in the following section, or numerically. For a total number of $M$ time steps in a trajectory, the (normalized) mean activity $\tilde{A}(\phi)$ as a function of some head direction $\phi$ is then

$$\tilde{A}(\phi) = \frac{1}{M} \sum_{m=0}^{M-1} \delta(\phi - \theta_m)\, \bar{A}(t_m). \tag{10}$$

where $\delta$ is the Dirac delta distribution. With complex Fourier coefficients $c_n$ (with $n \in \mathbb{N}$) defined as

$$c_n = \int_0^{2\pi} d\phi\, c(\phi)\, \exp(-nj\phi) \tag{11}$$

we can quantify the hexasymmetry $H$ of the activity of a population of grid cells as

$$H := \left| \tilde{A}_6 \right| = \int_0^{2\pi} d\phi\, \tilde{A}(\phi)\, \exp(-6j\phi) = \left| \frac{1}{M} \sum_{m=0}^{M-1} \bar{A}(t_m) \exp(-6j\theta_m) \right| = \left| \sum_{i=1}^{N} \tilde{A}_{i6} \right| \tag{12}$$

where $\tilde{A}_{i6}$ is the sixth Fourier coefficient of cell $i$. The hexasymmetry is nonnegative ($H \geq 0$) and it has the unit spk/s (spikes per second). Furthermore, the average (over time) population activity can be expressed as

$$\tilde{A}_0 = \frac{1}{M} \sum_{m=0}^{M-1} \bar{A}(t_m) . \tag{13}$$

The hexasymmetry $H$ could be generated by various properties of the cells, but $H$ as defined above contains also contributions from the hexasymmetry of the underlying trajectory. This is due to the fact that we sum up in *Equation 10* the population activities $\bar{A}(t_m)$ without taking into account the distribution of movement directions $\theta_m$. In this way, a hexasymmetry that is potentially contained in the subject's navigation trajectory contributes to the hexasymmetry $H$ of the neural activity.

Empirical (fMRI/iEEG) studies (e.g. *Doeller et al., 2010*; *Kunz et al., 2015*; *Maidenbaum et al., 2018*) addressed this problem of trajectories spuriously contributing to hexasymmetry by fitting a generalized linear model (GLM) to the time-discrete fMRI/EEG activity. In contrast, our new approach to hexasymmetry in *Equation 12* quantifies the contribution of the path to the neural hexasymmetry explicitly, and has the advantage that it allows an analytical treatment (see next section). Comparing our new method with previous methods for evaluating hexasymmetry led to qualitatively identical statistical effects (*Figure 5—figure supplement 4*).

To nevertheless be able to quantify how much a specific trajectory contributed to the neural hexasymmetry $H = \left| \tilde{A}_6 \right|$, we also explicitly quantified the hexasymmetry of navigation trajectories and interpreted $H$ relative to the path hexasymmetry. To quantify the hexasymmetry of a trajectory, we used the same approach as in *Equation 10* and defined the distribution of movement directions of the trajectory by

$$\tilde{T}(\phi) = \frac{1}{M} \sum_{m=0}^{M-1} \delta(\phi - \theta_m). \tag{14}$$

The path hexasymmetry of the trajectory is then $|\tilde{T}_6| := \left| \frac{1}{M} \sum_{m=0}^{M-1} \exp(-6j\theta_m) \right|$, which is similar to *Equation 12*. The path hexasymmetry is a unitless quantity, and ranges from 0 (for a uniform sampling of movement directions) to 1 (when only one movement direction is sampled).

To be able to estimate how much of the hexasymmetry $H$ of neuronal activity is due to the hexasymmetry of the trajectory, we compare the relative hexasymmetry of the activity, $|\tilde{A}_6/\tilde{A}_0|$, with the relative hexasymmetry of the trajectory, $|\tilde{T}_6/\tilde{T}_0|$, noting that $\tilde{T}_0 \equiv 1$. The two terms being similar in magnitude, i.e., $|\tilde{A}_6/\tilde{A}_0| \approx |\tilde{T}_6|$, indicates that the trajectory is a major source of hexasymmetry whereas $H = |\tilde{A}_6| \gg \tilde{A}_0 |\tilde{T}_6|$ suggests that hexasymmetry has a neural origin.

## Analytical derivation of mean activity

In the following, we provide derivations that allow us to analytically integrate *Equation 9* for the conjunctive grid by HD cell hypothesis and the structure-function mapping hypothesis but not for the repetition suppression hypothesis. The respective results are shown in *Figure 2B–H* and *Figure 4A–C*, demonstrating that they are very similar to the numerical results.

We analytically calculate the mean activity $\bar{A}$ by averaging $A$ along a linear segment of a trajectory (*Equation 9*). For convenience, the following abbreviations are used in *Equation 2* with the same grid spacing, $s_i = s$, and the same grid orientation, $\gamma_i = \gamma$, for all cells $i$:

$$a := \frac{4\pi}{\sqrt{3}s} \tag{15}$$

$$b_{x,i} := s\,x_{\text{off},i}, \; b_{y,i} := s\,y_{\text{off},i} \tag{16}$$

$$c_{k,x} := \sin\left(\frac{\pi}{3}k + \gamma\right), \; c_{k,y} := \cos\left(\frac{\pi}{3}k + \gamma\right). \tag{17}$$

### A single grid cell

We start with a single grid cell without HD tuning and without repetition suppression. *Equation 2* can be described in polar coordinates

$$G_i(r_t, \psi_t) = \frac{A_{\max}}{8} \prod_{k=0}^{2} \left(1 + \cos\left[a\,c_{k,x}\left(r_t\cos(\psi_t) - b_{x,i}\right) + a\,c_{k,y}\left(r_t\sin(\psi_t) - b_{y,i}\right)\right]\right). \tag{18}$$

In order to integrate along a piece of a straight line through the origin (similarly to the star-like walk), the angle $\psi_t \equiv \bar{\psi}$ can now be kept fixed (for that particular straight line) and we only have to consider $G_i(r_t, \bar{\psi})$. If we define $r_m$ and $r_{m+1}$ as the distances from zero that the subject is located at times $t_m$ and $t_{m+1}$ respectively, integration by substitution gives us

$$\int_{t_m}^{t_{m+1}} dt\, r'_t\, G_i(r_t, \bar{\psi}) = \int_{r_m}^{r_{m+1}} dr\, G_i(r, \bar{\psi}). \tag{19}$$

Since the speed of movement $r'_t \equiv v$ is assumed to be constant along the whole trajectory and $\Delta r = v\Delta t$, for $\Delta r := r_{m+1} - r_m$, we obtain

$$\frac{1}{\Delta t} \int_{t_m}^{t_{m+1}} dt\, G_i(r_t, \bar{\psi}) = \frac{1}{\Delta t} \cdot \frac{1}{v} \int_{r_m}^{r_{m+1}} dr\, G_i(r, \bar{\psi}) = \frac{1}{\Delta r} \int_{r_m}^{r_{m+1}} dr\, G_i(r, \bar{\psi}). \tag{20}$$

Thus, we have

$$\bar{A}(t_m) = \frac{1}{\Delta r} \int_{r_m}^{r_{m+1}} dr\, G_i(r, \bar{\psi}) = \tag{21}$$

$$= \frac{1}{\Delta r} \frac{A_{\max}}{8} \int_{r_m}^{r_{m+1}} dr \prod_{k=0}^{2} \left(1 + \cos\left[a\,c_{k,x}\left(r\cos(\bar{\psi}) - b_{x,i}\right) + a\,c_{k,y}\left(r\sin(\bar{\psi}) - b_{y,i}\right)\right]\right) = \tag{22}$$

$$= \frac{1}{\Delta r} \frac{A_{\max}}{8} \left( \underbrace{\int_{r_m}^{r_{m+1}} dr\, 1}_{=:(A)} + \sum_{k=0}^{2} \underbrace{\int_{r_m}^{r_{m+1}} dr\, z_k(r)}_{=:(B_k)} + \underbrace{\int_{r_m}^{r_{m+1}} dr\, z_0(r)z_1(r)}_{=:(C_{0,1})} \right. \tag{23}$$

$$\left. + \underbrace{\int_{r_m}^{r_{m+1}} dr\, z_0(r)z_2(r)}_{=:(C_{0,2})} + \underbrace{\int_{r_m}^{r_{m+1}} dr\, z_1(r)z_2(r)}_{=:(C_{1,2})} + \underbrace{\int_{r_m}^{r_{m+1}} dr\, z_0(r)z_1(r)z_2(r)}_{=:(D)} \right), \tag{24}$$

where $z_k(r) := \cos\left[a\,c_{k,x}\left(r\cos(\bar{\psi}) - b_{x,i}\right) + a\,c_{k,y}\left(r\sin(\bar{\psi}) - b_{y,i}\right)\right]$. The four parts (A)–(D) are integrated separately. We obtain

$$(A) = r\Big|_{r_m}^{r_{m+1}} \tag{25}$$

$$(B_k) = \begin{cases} \sin(r\,d_k - e_k) \cdot \dfrac{1}{d_k} \Big|_{r_m}^{r_{m+1}} & \text{if } d_k \neq 0 \\[2mm] \cos(e_k)r \Big|_{r_m}^{r_{m+1}} & \text{if } d_k = 0 \end{cases} \tag{26}$$

$$(C_{k,l}) = \begin{cases} \dfrac{1}{2}\left[\sin(r(d_k + d_l) - (e_k + e_l))\dfrac{1}{d_k + d_l} + \sin(r(d_k - d_l) - (e_k - e_l))\dfrac{1}{d_k - d_l}\right]\Big|_{r_m}^{r_{m+1}} & \text{if } d_k \neq \pm d_l \\[2mm] \left[\sin(2r\,d_k - e_k - e_l) + 2r\,d_k \cos(e_k - e_l)\right] \cdot \dfrac{1}{4d_k}\Big|_{r_m}^{r_{m+1}} & \text{if } d_k = d_l \neq 0 \\[2mm] \left[\sin(2r\,d_k - e_k + e_l) + 2r\,d_k \cos(e_k + e_l)\right] \cdot \dfrac{1}{4d_k}\Big|_{r_m}^{r_{m+1}} & \text{if } d_k = -d_l \neq 0 \\[2mm] r\cos(e_k)\cos(e_l)\Big|_{r_m}^{r_{m+1}} & \text{if } d_k = d_l = 0 \end{cases} \tag{27}$$

$$(D) = \frac{1}{4}\sum_{\epsilon_1,\epsilon_2 \in \{-1,1\}}\left[\left(1 - [[d_0 + \epsilon_1 d_1 + \epsilon_2 d_2 = 0]]\right)\sin\left((d_0 + \epsilon_1 d_1 + \epsilon_2 d_2)r - (e_0 + \epsilon_1 e_1 + \epsilon_2 e_2)\right)\frac{1}{d_0 + \epsilon_1 d_1 + \epsilon_2 d_2}\right.$$
$$\left. + [[d_0 + \epsilon_1 d_1 + \epsilon_2 d_2 = 0]]\cos(e_0 + \epsilon_1 e_1 + \epsilon_2 e_2)r \right]\Big|_{r_m}^{r_{m+1}} \tag{28}$$

where $[[Q]] := \begin{cases} 1 & \text{if } Q \text{ is true} \\ 0 & \text{else} \end{cases}$ is the Iverson bracket, and with the abbreviations

$$d_k = a\left(c_{k,x}\cos(\bar{\psi}) + c_{k,y}\sin(\bar{\psi})\right) \tag{29}$$

$$e_k = a\left(c_{k,x}\,b_{x,i} + c_{k,y}\,b_{y,i}\right). \tag{30}$$

Note that $e_k$ actually depends on the cell index $i$ which is omitted in **Equation 25** and **Equation 28** in order to keep the notation simpler.

## HD tuning

For a conjunctive grid by HD cell, the HD tuning depends only on the angle (which is fixed when integrating along a straight line through zero) and not at all on the distance from zero. The mean activity is thus obtained from the mean activity of a grid cell without HD tuning by multiplying it with $h(\theta)$ defined in **Equation 4**:

$$\frac{1}{\Delta r}\int_{r_m}^{r_{m+1}} dr\, G_i(r, \bar{\psi})h_i(\bar{\psi}) = \frac{1}{\Delta r}\,h_i(\bar{\psi})\int_{r_m}^{r_{m+1}} dr\, G_i(r, \bar{\psi}). \tag{31}$$

## Many cells

For more than one cell, the total mean activity (population rate) can simply be calculated as the sum of the mean activities of the single cells

$$\frac{1}{\Delta r}\int_{r_m}^{r_{m+1}} dr\,\sum_{i=1}^{N} G_i(r, \bar{\psi})h_i(\bar{\psi}) = \frac{1}{\Delta r}\sum_{i=1}^{N} h_i(\bar{\psi})\int_{r_m}^{r_{m+1}} dr\, G_i(r, \bar{\psi}). \tag{32}$$

## Trajectories

The derived analytical description of the mean activity can be applied only to pieces of linear trajectories through zero. For the star-like walk, we can simply integrate

$$\frac{1}{r_{\max}}\int_{0}^{r_{\max}} dr\, G_i(r, \theta) \quad \text{for } \theta \in \left\{0, \frac{2\pi}{N_\theta}, 2\frac{2\pi}{N_\theta}\cdots\right\}. \tag{33}$$

Piecewise linear trajectories and random-walk trajectories consist of segments of straight lines that do not necessarily pass through zero. In order to integrate along the $m$-th segment of a trajectory from $(x_{t_m}, y_{t_m})$ to $(x_{t_{m+1}}, y_{t_{m+1}})$ with movement direction $\theta_m$, we shift this path segment to the origin by subtracting $(x_{t_m}, y_{t_m})$ from the grid offset $(x_{\text{off}}, y_{\text{off}})$ and then integrating

$$\frac{1}{r_{m+1}}\int_{0}^{r_{m+1}} dr\, G_i(r, \theta_m), \tag{34}$$

where $r_{m+1} = \sqrt{(x_{t_{m+1}} - x_{t_m})^2 + (y_{t_{m+1}} - y_{t_m})^2}$.

## Upper bound for hexasymmetry of path trajectories

In the following we derive approximations for the expected value of the hexasymmetry $|\tilde{T}_6|$ of a path trajectory described by the number of time steps $M$, the movement tortuosity $\sigma_\theta$, and the time step size $\Delta t$. These approximations can be used to assess the contribution of the underlying trajectory to the overall hexasymmetry of neural activity.

From the definition of the Fourier coefficients in *Equation 11* and the movement direction distribution in *Equation 14*, we get the sixth Fourier coefficient of a trajectory:

$$
\begin{aligned}
\tilde{T}_6 &= \int_0^{2\pi} \mathrm{d}\phi \, \frac{1}{M} \sum_{m=0}^{M-1} \delta(\phi - \theta_m) \exp(-6j\phi) \\
&= \frac{1}{M} \sum_{m=0}^{M-1} \exp(-6j\theta_m) = \frac{1}{M} \left( \sum_{m=0}^{M-1} \cos(6\,\theta_m) - j \sum_{m=0}^{M-1} \sin(6\,\theta_m) \right).
\end{aligned}
\tag{35}
$$

The hexasymmetry of the trajectory can thus be expressed as

$$
|\tilde{T}_6| = \frac{1}{M} \sqrt{\left( \sum_{m=0}^{M-1} \cos(6\,\theta_m) \right)^2 + \left( \sum_{m=0}^{M-1} \sin(6\,\theta_m) \right)^2}.
\tag{36}
$$

We simplify the sum of squares in *Equation 36*

$$
\begin{aligned}
\left( \sum_{m=0}^{M-1} \cos(6\,\theta_m) \right)^2 + \left( \sum_{m=0}^{M-1} \sin(6\,\theta_m) \right)^2 &= \sum_{m=0}^{M-1} (\cos(6\,\theta_m))^2 + 2 \sum_{m_1=0}^{M-1} \sum_{m_2=m_1+1}^{M-1} \cos(6\,\theta_{m_1}) \cos(6\,\theta_{m_2}) \\
&+ \sum_{m=0}^{M-1} (\sin(6\,\theta_m))^2 + 2 \sum_{m_1=0}^{M-1} \sum_{m_2=m_1+1}^{M-1} \sin(6\,\theta_{m_1}) \sin(6\,\theta_{m_2}) \\
&= M + \sum_{m_1=0}^{M-1} \sum_{\substack{m_2=0 \\ m_2 \neq m_1}}^{M-1} \cos(6\,(\theta_{m_1} - \theta_{m_2}))
\end{aligned}
\tag{37}
$$

with the help of the multinomial theorem and the trigonometric identity $\cos(\alpha - \beta) = \cos(\alpha)\cos(\beta) + \sin(\alpha)\sin(\beta)$. If $\mathbb{E}\left(\cos(6(\theta_{m_1} - \theta_{m_2}))\right)$ is known, we can compute the expected value of the square of the hexasymmetry of the path trajectory as

$$
\begin{aligned}
\mathbb{E}\left( |\tilde{T}_6|^2 \right) &= \mathbb{E}\left( \frac{1}{M^2} \left( M + \sum_{m_1=0}^{M-1} \sum_{\substack{m_2=0 \\ m_2 \neq m_1}}^{M-1} \cos(6\,(\theta_{m_1} - \theta_{m_2})) \right) \right) \\
&= \frac{1}{M^2} \left( M + \sum_{m_1=0}^{M-1} \sum_{\substack{m_2=0 \\ m_2 \neq m_1}}^{M-1} \mathbb{E}\left( \cos(6(\theta_{m_1} - \theta_{m_2})) \right) \right)
\end{aligned}
\tag{38}
$$

As $\mathbb{E}(X^2) = (\mathbb{E}(X))^2 + \mathrm{Var}(X)$ for any random variable $X$, we can use the result in *Equation 38* to obtain the upper bound

$$
\mathbb{E}\left( |\tilde{T}_6| \right) \leq \sqrt{\mathbb{E}\left( |\tilde{T}_6|^2 \right)}.
\tag{39}
$$

In the following, we focus on the derivation of $\mathbb{E}\left(\cos(6(\theta_{m_1} - \theta_{m_2}))\right)$. Using the movement statistics

$$
\Delta W \sim \mathcal{N}(0, \Delta t)
\tag{40}
$$

$$
\theta_{t+\Delta t} = \theta_t + \sigma_\theta \Delta W
\tag{41}
$$

that were introduced below *Equation 1*, the distribution of $\theta_m$ (after $m$ time steps) can be derived when we start at some angle $\theta_0$:

$$\theta_1 = \theta_0 + \sigma_\theta \Delta W \sim \mathcal{WN}\left(\theta_0, \sigma_\theta^2 \Delta t\right),$$

$$\theta_2 = \theta_1 + \sigma_\theta \Delta W \sim \mathcal{WN}\left(\theta_0, 2\sigma_\theta^2 \Delta t\right)$$

$$\vdots$$

$$\theta_m \sim \mathcal{WN}\left(\theta_0, m\sigma_\theta^2 \Delta t\right) \tag{42}$$

where $\mathcal{WN}(\mu, \sigma^2)$ denotes the wrapped normal distribution with parameters $\mu$ and $\sigma^2$, which correspond to the mean and variance of the corresponding unwrapped distribution (*Jammalamadaka and SenGupta, 2001*).

In the following, we will derive the probability distribution of $\theta_{m_1} - \theta_{m_2}$. We first define $X_{m_1} \sim \mathcal{N}(\theta_0, m_1\sigma_\theta^2 \Delta t)$ and $X_{m_2} \sim \mathcal{N}(\theta_0, m_2\sigma_\theta^2 \Delta t)$ as the unwrapped versions of $\theta_{m_1}$ and $\theta_{m_2}$, respectively, and $G_{X_{m_1} - X_{m_2}}$ as the distribution function of their difference $X_{m_1} - X_{m_2}$. Then, the distribution function (*Fisher, 1995*) of $\theta_{m_1} - \theta_{m_2}$ reads as

$$F_{\theta_{m_1} - \theta_{m_2}}(z) = \sum_{k=-\infty}^{\infty} \left[ G_{X_{m_1} - X_{m_2}}(z + 2\pi k) - G_{X_{m_1} - X_{m_2}}(2\pi k) \right) \tag{43}$$

$$= \sum_{k=-\infty}^{\infty} \left[ \iint_{x-y \leq z+2\pi k} f_{X_{m_1}, X_{m_2}}(x, y) \mathrm{d}x\,\mathrm{d}y - \iint_{x-y \leq 2\pi k} f_{X_{m_1}, X_{m_2}}(x, y) \mathrm{d}x\,\mathrm{d}y \right] \tag{44}$$

$$= \sum_{k=-\infty}^{\infty} \left[ \int_{-\infty}^{\infty} \int_{-\infty}^{z+2\pi k+y} f_{X_{m_1}, X_{m_2}}(x, y) \mathrm{d}x\,\mathrm{d}y - \int_{-\infty}^{\infty} \int_{-\infty}^{2\pi k+y} f_{X_{m_1}, X_{m_2}}(x, y) \mathrm{d}x\,\mathrm{d}y \right] \tag{45}$$

$$= \sum_{k=-\infty}^{\infty} \int_{-\infty}^{\infty} \int_{2\pi k+y}^{z+2\pi k+y} f_{X_{m_1}, X_{m_2}}(x, y)\,\mathrm{d}x\,\mathrm{d}y \tag{46}$$

where $f_{X_{m_1}, X_{m_2}}(x, y)$ is the joint probability distribution.

In order to calculate the distribution of $\theta_{m_1} - \theta_{m_2}$, we have to take the dependence between the two angles $\theta_{m_1}$ and $\theta_{m_2}$ into account. We will first consider the case $m_1 > m_2$. In this case, the conditional distribution of $\theta_{m_1}$ given $\theta_{m_2} = y$ is wrapped normal with conditional mean $\mathbb{E}(\theta_{m_1}|\theta_{m_2} = y) = y$ and conditional variance $\mathrm{Var}(\theta_{m_1}|\theta_{m_2} = y) = (m_1 - m_2)\sigma_\theta^2 \Delta t$. The same can be said about the unwrapped versions of $\theta_{m_1}$ and $\theta_{m_2}$. We will use this result to calculate the joint probability distribution

$$f_{X_{m_1}, X_{m_2}}(x, y) = f_{X_{m_1}}(x|X_{m_2} = y) f_{X_{m_2}}(y). \tag{47}$$

By applying the Leibniz integral rule in $(*)$ we obtain the probability density of $\theta_{m_1} - \theta_{m_2}$:

$$f_{\theta_{m_1} - \theta_{m_2}}(z) = \frac{\mathrm{d}}{\mathrm{d}z} F_{\theta_{m_1} - \theta_{m_2}}(z) = \frac{\mathrm{d}}{\mathrm{d}z} \sum_{k=-\infty}^{\infty} \int_{-\infty}^{\infty} \int_{2\pi k+y}^{z+2\pi k+y} f_{X_{m_1}, X_{m_2}}(x, y)\,\mathrm{d}x\,\mathrm{d}y \tag{48}$$

$$\overset{(*)}{=} \sum_{k=-\infty}^{\infty} \int_{-\infty}^{\infty} f_{X_{m_1}, X_{m_2}}(z + 2\pi k + y, y)\,\mathrm{d}y \tag{49}$$

$$= \sum_{k=-\infty}^{\infty} \int_{-\infty}^{\infty} f_{X_{m_1}}(z + 2\pi k + y|X_{m_2} = y) f_{X_{m_2}}(y)\,\mathrm{d}y \tag{50}$$

$$= \sum_{k=-\infty}^{\infty} \int_{-\infty}^{\infty} \frac{1}{\sigma_\theta \sqrt{2\pi(m_1 - m_2)\Delta t}} \exp\left( -\frac{(z + 2\pi k + y - y)^2}{2(m_1 - m_2)\sigma_\theta^2 \Delta t} \right) \tag{51}$$

$$\cdot \frac{1}{\sigma_\theta \sqrt{2\pi m_2 \Delta t}} \exp\left( -\frac{(y - \theta_0)^2}{2m_2 \sigma_\theta^2 \Delta t} \right) \mathrm{d}y \tag{52}$$

$$= \sum_{k=-\infty}^{\infty} \left[ \frac{1}{\sigma_\theta \sqrt{2\pi(m_1 - m_2)\Delta t}} \exp\left( -\frac{(z + 2\pi k)^2}{2(m_1 - m_2)\sigma_\theta^2 \Delta t} \right) \right] \cdot \underbrace{\int_{-\infty}^{\infty} f_{X_{m_2}}(y)\,\mathrm{d}y}_{=1}. \tag{53}$$

For $m_2 > m_1$, we derive in the same way as above

$$f_{\theta_{m_1} - \theta_{m_2}}(z) = \sum_{k=-\infty}^{\infty} \int_{-\infty}^{\infty} f_{X_{m_1}}(z + 2\pi k + y) f_{X_{m_2}}(y|X_{m_1} = z + 2\pi k + y)\, dy$$

$$= \sum_{k=-\infty}^{\infty} \frac{1}{\sigma_\theta \sqrt{2\pi(m_2 - m_1)\Delta t}} \exp\left(-\frac{(z + 2\pi k)^2}{2(m_2 - m_1)\sigma_\theta^2 \Delta t}\right). \tag{54}$$

Finally, for $m_2 = m_1$, we have $\theta_{m_2} = \theta_{m_1}$. Altogether, we thus get

$$\theta_{m_1} - \theta_{m_2} \sim \mathcal{WN}\left(0, |m_1 - m_2|\sigma_\theta^2 \Delta t\right). \tag{55}$$

In order to calculate an upper bound for the average path hexasymmetry, we will now use *Equation 55* in *Equation 38*. Since $\mathbb{E}(\Re(Z)) = \Re(\mathbb{E}(Z))$ for a complex random variable $Z$ and the $n$ th moment of a wrapped normal distribution with parameters $\mu$ and $\sigma^2$ is $\mathbb{E}(\exp(jX)^n) = \exp(jn\mu - \frac{1}{2}n^2\sigma^2)$, we can derive

$$\mathbb{E}\left(\cos(6(\theta_{m_1} - \theta_{m_2}))\right) = \mathbb{E}\left(\Re\left(\exp(j6(\theta_{m_1} - \theta_{m_2}))\right)\right) = \Re\left(\mathbb{E}\left(\exp(j(\theta_{m_1} - \theta_{m_2}))^6\right)\right)$$

$$= \Re\left(\exp\left(j6\mu_{m_1,m_2} - \frac{1}{2}36\sigma_{m_1,m_2}^2\right)\right) = \exp\left(-\frac{1}{2}36\sigma_{m_1,m_2}^2\right), \tag{56}$$

where $\mu_{m_1,m_2} := 0$ and $\sigma_{m_1,m_2}^2 := |m_1 - m_2|\sigma_\theta^2 \Delta t$. We thus obtain

$$\mathbb{E}\left(|\tilde{T}_6|^2\right) = \frac{1}{M^2}\left(M + \sum_{m_1=0}^{M-1}\sum_{\substack{m_2=0 \\ m_2 \neq m_1}}^{M-1} \exp\left(-\frac{1}{2}36\sigma_{m_1,m_2}^2\right)\right)$$

$$= \frac{1}{M^2}\left(M + \sum_{m_1=0}^{M-1}\sum_{\substack{m_2=0 \\ m_2 \neq m_1}}^{M-1} \exp\left(-\frac{1}{2}36|m_1 - m_2|\sigma_\theta^2 \Delta t\right)\right). \tag{57}$$

The solid lines in *Figure 1—figure supplement 4* show the square root of *Equation 57* (*Equation 39*).

From *Equation 57* we can derive simplified approximations for two limiting cases. For convenience, we use $\alpha := \frac{1}{2}36\sigma_\theta^2 \Delta t$. First, if $\alpha \gg 1$, i.e., if the new direction after one step is almost uniformly distributed or independent of the previous direction, we can neglect the double sum and we have

$$\mathbb{E}\left(|\tilde{T}_6|\right) \leq \frac{1}{\sqrt{M}}. \tag{58}$$

The corresponding line is shown in red in *Figure 1—figure supplement 4*. Note that in *Figure 1—figure supplement 4* (with simulation step size $\Delta t = 0.01$ s), $\alpha \gg 1$ corresponds to $\sigma_\theta \gg 2.36$. A comparison with results from numerical simulations shows that for any $\sigma_\theta > 3.5$ *Equation 58* constitutes a viable upper bound of the mean path hexasymmetry.

Second, we assume $M\alpha \gg 1$, i.e., the direction after $M$ steps is almost independent from the original direction. The double sum in *Equation 57* can then be approximated:

$$\sum_{m_1=0}^{M-1}\sum_{\substack{m_2=0 \\ m_2 \neq m_1}}^{M-1} \exp\left(-\alpha|m_1 - m_2|\right) = 2\sum_{m_1=0}^{M-1}\sum_{m_2=m_1+1}^{M-1} \exp\left(-\alpha|m_1 - m_2|\right) \tag{59}$$

$$= 2\sum_{m=1}^{M-1}(M - m)\exp(-\alpha m) \approx 2\sum_{m=1}^{\infty}(M - m)\exp(-\alpha m) \tag{60}$$

$$= 2\left(M\left(\sum_{m=0}^{\infty}(\exp(-\alpha))^m - 1\right) - \sum_{m=1}^{\infty}m(\exp(-\alpha))^m\right) \tag{61}$$

$$= 2\left(\frac{M}{\exp(\alpha) - 1} - \frac{\exp(\alpha)}{(\exp(\alpha) - 1)^2}\right), \tag{62}$$

where the first series in *Equation 61* is a geometric series and the second series is the polylogarithm function of order –1. We further approximate

$$2 \left( \frac{M}{\exp(\alpha) - 1} - \frac{\exp(\alpha)}{\left(\exp(\alpha) - 1\right)^2} \right) \approx \frac{2M}{\exp(\alpha) - 1}. \tag{63}$$

For $\alpha \ll 1$, the error in the last approximation is $\frac{\exp(\alpha)}{\left(\exp(\alpha) - 1\right)^2} \approx \frac{1+\alpha}{\alpha^2}$, which, if $M\alpha \gg 1$, is negligible compared to the remaining term $\frac{M}{\exp(\alpha) - 1}$. Since $\frac{\exp(\alpha)}{\exp(\alpha) - 1}$ is a strictly monotonically decreasing function of $\alpha$, this approximation does not only hold for $\alpha \ll 1$ but is good in general.

Inserting *Equation 63* into *Equation 57* gives

$$\mathbb{E}\left(\left|\tilde{T}_6\right|^2\right) = \frac{1}{M^2}\left(M + \frac{2M}{\exp(\alpha) - 1}\right) = \frac{1}{M}\left(1 + \frac{2}{\exp(\alpha) - 1}\right). \tag{64}$$

Hence, we get the approximation (for $M\alpha \gg 1$)

$$\mathbb{E}\left(\left|\tilde{T}_6\right|\right) \lesssim \sqrt{\frac{1}{M}\left(1 + \frac{2}{\exp(\alpha) - 1}\right)}. \tag{65}$$

This expression is used to compute the dashed lines in *Figure 1—figure supplement 4*, which all have slope $1/\sqrt{M}$ but a prefactor that depends on $\alpha = 18\sigma_\theta^2 \Delta t$.

For $\alpha \ll 1$ (but still $M\alpha \gg 1$), we can use in *Equation 64* the first-order Taylor expansion of the exponential function at 0 to obtain

$$\frac{1}{M}\left(1 + \frac{2}{\exp(\alpha) - 1}\right) \approx \frac{1}{M}\left(1 + \frac{2}{\alpha}\right) \approx \frac{1}{M} \cdot \frac{2}{\alpha} = \frac{1}{M} \cdot \frac{1}{9\sigma_\theta^2 \Delta t}. \tag{66}$$

Hence, the path hexasymmetry for $\alpha \ll 1$ and $M\alpha \gg 1$ can be approximated by

$$\mathbb{E}\left(\left|\tilde{T}_6\right|\right) \lesssim \frac{1}{\sqrt{M}} \cdot \frac{1}{3\sigma_\theta \sqrt{\Delta t}}, \tag{67}$$

which allows us to see how the key variables $M$, $\sigma_\theta$, and $\Delta t$ interact in this limiting case. For instance, given a certain trajectory $A$ with $M_A$ steps and random-walk parameters $\sigma_{\theta A}$ and $\Delta t_A$, we can use *Equation 67* to derive how many steps $M_B$ are necessary in a second trajectory $B$ with parameters $\sigma_{\theta B}$ and $\Delta t_B$ to achieve the same mean path hexasymmetry. From *Equation 67*, we know that the two path hexasymmetries will have the same upper bound if

$$\frac{1}{\sqrt{M_A}} \cdot \frac{1}{3\sigma_{\theta A}\sqrt{\Delta t_A}} = \frac{1}{\sqrt{M_B}} \cdot \frac{1}{3\sigma_{\theta B}\sqrt{\Delta t_B}} \quad \Leftrightarrow \quad \frac{M_B}{M_A} = \frac{\sigma_{\theta A}^2 \Delta t_A}{\sigma_{\theta B}^2 \Delta t_B}. \tag{68}$$

Hence, the number of time steps $M_A$ has to be multiplied by a factor $\Delta M := \frac{\sigma_{\theta A}^2 \Delta t_A}{\sigma_{\theta B}^2 \Delta t_B}$:

$$M_B = \Delta M \cdot M_A. \tag{69}$$

We illustrate the above considerations with an example: Let $\sigma_{\theta A} = 1\,\mathrm{rad/s}^{1/2}$, $\sigma_{\theta B} = 0.5\,\mathrm{rad/s}^{1/2}$, and $\Delta t_A = \Delta t_B = 0.01$ s. From the given values, we obtain

$$\Delta M = \frac{\sigma_{\theta A}^2 \Delta t_A}{\sigma_{\theta B}^2 \Delta t_B} = \frac{1}{0.25} = 4. \tag{70}$$

Thus, the mean hexasymmetry value of a trajectory with $\sigma_{\theta A} = 1$ after $M_A$ time steps is the same as the mean hexasymmetry value of a trajectory with $\sigma_{\theta B} = 0.5$ after $M_B = 4 \cdot M_A$ time steps. Results from numerical simulations of path hexasymmetries, shown in *Figure 1—figure supplement 4*, support the derived theoretical approximations.

## Random-field simulations

To quantitatively evaluate the structure-function mapping hypothesis, we set out to simulate a set of grid cells in three-dimensional anatomical space. The grid phases associated with these grid cells follow the correlation structure suggested by *Gu et al., 2018*, and *Heys et al., 2014*. Our aim is to quantify the clustering of grid phases for a realistically sized fMRI voxel given this correlation structure.

We use a three-dimensional representation of a voxel with a volume of $3 \times 3 \times 3$ mm³. Within this voxel, we define a grating of $200^3$ potential grid cells that are equally spaced in the voxel, with a distance between neighboring cells of 15 μm along the axes of the grating. To generate a set of random but spatially correlated grid phases on this area, we first define two random unit vectors in the complex plane, $\mathbf{Z}_1$ and $\mathbf{Z}_2$, for each of the $200^3$ potential grid cells in the voxel; angles of the unit vectors are thus drawn from a uniform distribution on the interval $[0, 2\pi)$. $\mathbf{Z}_1$ and $\mathbf{Z}_2$ are further resolved into their real and imaginary components $\text{Re}(\mathbf{Z}_i)$ and $\text{Im}(\mathbf{Z}_i)$, respectively, where $i \in \{1, 2\}$. To generate correlations between grid phases, we then convolve the two resulting gratings of $200^3$ components separately with either a Gaussian kernel (*Figure 4K*) or a grid kernel (*Figure 4O*) to yield the convolved components $\text{Re}(\hat{\mathbf{Z}}_i)$ and $\text{Im}(\hat{\mathbf{Z}}_i)$. The grid phases can be obtained by first calculating the angles of the new set of complex numbers and normalizing the result by $2\pi$:

$$\begin{aligned} \hat{x}_{\text{off}} &= \frac{\arg(\hat{\mathbf{Z}}_1)}{2\pi} \\ \hat{y}_{\text{off}} &= \frac{\arg(\hat{\mathbf{Z}}_2)}{2\pi}. \end{aligned} \tag{71}$$

We note that $\hat{x}_{\text{off}}$ and $\hat{y}_{\text{off}}$ are defined on the interval $[0, 1)$ and correspond to the grid phases of a single grid cell mapped to the unit square. Transforming the result to the unit rhombus yields the grid phases $x_{\text{off}}$ and $y_{\text{off}}$ in the $x$ and $y$ direction respectively:

$$\begin{aligned} x_{\text{off}} &= \hat{x}_{\text{off}} + \frac{\hat{y}_{\text{off}}}{2} \\ y_{\text{off}} &= \frac{\sqrt{3}}{2}\hat{y}_{\text{off}} \end{aligned} \tag{72}$$

To find the average pairwise grid phase distances as a function of the pairwise anatomical distances, $10^8$ pairs of grid cells are sampled randomly from the uniform distribution defined on the discrete space of grating cell positions. The Euclidean distance in anatomical space between the two grid cells in each pair is calculated and sorted into 50 bins of equal width on the interval $[10, 500]$ μm. Then, for each pair of grid cells, $n_1$ and $n_2$, eight copies of the grid phase $(x_{\text{off},2}, y_{\text{off},2})$ of the second cell $n_2$ are made, which are offset from the initial position of the grid phase such that they are positioned at the same phase within unit rhombi laid end-to-end on a $3 \times 3$ grid. The minimum distance between the grid phase of the cell $n_1$ and the grid phase of each of the copies of the cell $n_2$ is taken as the pairwise phase distance. Finally, the pairwise distance between grid phase offsets per distance bin is obtained by taking the mean over all grid cell pairs whose Euclidean distance in anatomical space falls into the corresponding bin (*Figure 4H, L, and P*).

To estimate the clustering concentration parameter $\kappa_s$ in *Figure 4*, the phases $\hat{x}_{\text{off}}$ and $\hat{y}_{\text{off}}$ are mapped to a circular distribution by multiplying them with $2\pi$. The sets of grid phases $\left\{ 2\pi\hat{x}_{\text{off}}^i \,|\, i \in \{1, 2, ..., N\} \right\}$ and $\left\{ 2\pi\hat{y}_{\text{off}}^i \,|\, i \in \{1, 2, ..., N\} \right\}$ are then each separately fit to a one-dimensional von Mises distribution to obtain a clustering concentration parameter for each axis. The final value of $\kappa_s$ is taken as the average of these two values.

## Parameter estimation

The hexasymmetry values of the tested hypotheses depend on the respective parameters. We investigate two scenarios for each hypothesis: an 'ideal' set of parameters that results in a large (basically maximal) hexasymmetry value and a 'realistic' set of parameters that we try to derive from experimental data. All sets of parameters are summarized in *Table 1*, i.e., the symbols are introduced and defined, and parameter values and units are stated. We comment here on the 'realistic' sets of parameters and how we derived them.

### Conjunctive grid by HD hypothesis

For the 'realistic' value of $\sigma_c = 3°$, we use the 95% confidence interval of 12.4° from the third row of the table in the Supplementary Figure 5b of *Doeller et al., 2010*, and translate this value to a Gaussian standard deviation, assuming a 95% confidence interval of $4\sigma_c$.

For the 'realistic' value of $\kappa_c = 4$, we refer to the Supplementary Figure 3 of *Doeller et al., 2010*. Visually, the tuning curves usually cover between one sixth and one third of the circle. Stronger HD

tuning contributes the most to the resulting hexasymmetry, so we choose a value of $\kappa_c = 4$, which corresponds to a tuning width of roughly one sixth of the circle. For further discussion on $\kappa_c$, see also *Figure 2—figure supplement 1*.

The 'realistic' value of $p_c = 1/3$ was chosen based on the values reported by *Sargolini et al., 2006*, and *Boccara et al., 2010*. In our simulations, we observed that the hexasymmetry shows a linear dependence on the fraction of conjunctive cells up to a noise floor (*Figure 2—figure supplement 2*).

## Repetition suppression hypothesis

To obtain 'realistic' parameter values, we simply divide the 'ideal' ones by two, which results in a 'realistic' adaptation time constant $\tau_r = 1.5$ s and a 'realistic' adaptation weight $w_r = 0.5$. 'Ideal' parameter values attenuate the firing rate by ≈20% (*Figure 3A*, with 24% when running aligned with the grid axes and 18% when running misaligned to the grid axes). 'Realistic' parameter values attenuate firing rates by ≈16% (for a dependence of hexasymmetry on the two parameters, see *Figure 3B*). Such a decrease has a similar order of magnitude as the attenuation of 5% of firing rates in *Reber et al., 2023* (their Figure 2B, right). The value of the 'realistic' time constant is also comparable to the time scales found in two electrophysiological studies: first, *Giocomo and Hasselmo, 2008*, showed that the time constant in the slow component of the hyperpolarization-activated current ($I_h$) in the ventral mEC is 2–3 s (their Figure 2F). Second, *Magistretti and Alonso, 1999*, found a second-long inactivation of a persistent sodium current in EC layer II cells.

## Structure-function mapping hypothesis

The 'realistic' value of the concentration parameter for clustering, $\kappa_s = 0.1$, is derived from the random-field simulations in *Figure 4G–S* and a comparison to results in *Gu et al., 2018*, and *Heys et al., 2014*. Figure 4G in *Gu et al., 2018*, shows for grid cells in the rat mEC the relationship between their pairwise physical distance and their pairwise phase distance. Similarly, for grid cells in the mouse mEC, *Heys et al., 2014*, in their Figure 4B show the correlation of (one-dimensional) grid firing maps as a function of anatomical pairwise distance between grid cells. Together, these data serve as evidence for more similar grid cell firing for anatomically close grid cells (distance <100 μm)—with decreasing similarity for increasing anatomical distance. In our random-field simulations (*Figure 4*), we mimic these experimental results by choosing the width of a spatial correlation kernel accordingly (*Figure 4K*, Gaussian kernel with standard deviation: 30 μm). Consequently, in the simulations, the initial rise of phase distance as a function of anatomical distance (*Figure 4L*) reflects the experimental curves. In Gu et al.'s Figure 4G there is a slight increase in grid-firing similarity (decrease of phase distance) for anatomical distances of approximately 300 μm. In our simulations, we reflect such a non-monotonous behavior by introducing a grid-like correlation kernel (*Figure 4O*) with a grid spacing of 300 μm. Applying this grid-like correlation kernel to the random field in *Figure 4G* results in a phase-distance curve with a slight dip around 300 μm (*Figure 4P*), just like in the experimental curves in *Gu et al., 2018*. For both cases, the Gaussian kernel and the grid-like kernel, we find that the resulting phase clustering for an anatomical voxel of realistic size is quite low, i.e., $\kappa_s$ is well below 0.1 (*Figure 4M and Q*). As it poses an upper bound for the realistic values of $\kappa_s$, we use 0.1 as an estimate of the 'realistic' value for $\kappa_s$ in our further simulations. For the central phase of the cluster, we always use (without loss of generality) the origin: $\mu_s = (0, 0)$.

## Implementation of previously used metrics

We applied three previously used metrics to our framework: the GLM method by *Doeller et al., 2010*; the GLM method with binning by *Kunz et al., 2015*, the circular-linear correlation method by *Maidenbaum et al., 2018*; *Figure 5—figure supplement 4*.

In brief, in the GLM method (e.g. used in *Doeller et al., 2010*), the hexasymmetry is found in two steps: the orientation of the hexadirectional modulation is first estimated on the first half of the data by using the regressors $\beta_1 \cos(6\theta_t)$ and $\beta_2 \sin(6\theta_t)$ on the time-discrete fMRI activity (*Equation 9*), with $\theta_t$ being the movement direction of the subject in time step $t$. The amplitude of the signal is then estimated on the second half of the data using the single regressor $\beta \cos[6(\theta_t - \phi)]$, where $\phi = \arctan(\beta_2/\beta_1)/6$. The hexasymmetry is then evaluated as $H = \beta/2$.

The GLM method with binning (e.g. used in *Kunz et al., 2015*) uses the same procedure as the GLM method for estimating the grid orientation in the first half of the data, but the amplitude is

estimated differently on the second half by a regressor that has a value 1 if $\theta_t$ is aligned with a peak of the hexadirectional modulation (aligned if $|\theta_t - \phi| \% 60° < 15°$, % modulo operator) and a value of $-1$ if $\theta_t$ is misaligned. The hexasymmetry is then calculated from the amplitude in the same way as the GLM method.

The circular-linear correlation method (e.g. used in *Maidenbaum et al., 2018*) is similar to the GLM method in that it uses the regressors $\beta_1 \cos(6\theta_t)$ and $\beta_2 \sin(6\theta_t)$ on the time-discrete mean activity, but instead of using $\beta_1$ and $\beta_2$ to estimate the orientation of the hexadirectional modulation, the beta values are directly used to estimate the hexasymmetry using the relation $H = \sqrt{\beta_1^2 + \beta_2^2}/2$.

Regarding the statistical evaluation, each method evaluates the size of the neural hexasymmetry differently. Specifically, the new method developed in our manuscript compares the neural hexasymmetry to path hexasymmetry to test whether neural hexasymmetry is significantly above path hexasymmetry. For the two GLM methods, we compare the hexasymmetry to zero (using the Mann-Whitney U test) to establish significance. Hexasymmetry values can be negative in these approaches, allowing the statistical comparison against 0. Negative values occur when the estimated grid orientation from the first data half does not match the grid orientation from the second data half. Regarding the statistical evaluation of the circular-linear correlation method, we calculated a z-score by comparing each empirical observation of the hexasymmetry to hexasymmetries from a set of surrogate distributions (as in *Maidenbaum et al., 2018*). We then calculate a p-value by comparing the distribution of z-scores versus zero using a Mann-Whitney U test. We use the z-scores instead of the hexasymmetry for the circular-linear correlation method to match the procedure used in *Maidenbaum et al., 2018*. We obtained the surrogate distributions by circularly shifting the vector of movement directions relative to the time-dependent vector of firing rates. For random walks, the vector is shifted by a random number drawn from a uniform distribution defined with the same length as the number of time points in the vector of movement directions. For the star-like walks and piecewise linear walks, the shift is a random integer multiplied by the number of time points in a linear segment. Circularly shifting the vector of movement directions scrambles the correlations between movement direction and neural activity while preserving their temporal structure.

### Tuning widths and the large-$\kappa$ approximation

The numerical values of our tuning width of HD modulated cells should not be directly compared to the numbers reported by *Sargolini et al., 2006*, with values in the range of 55° because *Sargolini et al., 2006* reported the tuning widths as the angular standard deviation $\sigma$ of the mean vector of HD-dependent firing rates whereas we derive the tuning width from the large-$\kappa_c$ approximation of a von Mises distribution: $\sigma = 1/\sqrt{\kappa_c}$. This simple approximation is reasonably good only for large enough $\kappa_c$ (or small enough $\sigma$), and 55° is already beyond the range where the approximation is reasonable (*Figure 2—figure supplement 1B*).

To examine the range of values where the large-$\kappa_c$ approximation holds, we can compare it to the angular standard deviation expressed as $\sigma = \sqrt{-2\ln[I_1(\kappa_c)/I_0(\kappa_c)]}$, where $I_n(\kappa_c)$ is the modified Bessel function of the first kind of order '$n$'. This equation is valid for all values of the concentration parameter $\kappa_c$. The results of this comparison are summarized in *Figure 2—figure supplement 1B*. In brief, the large-$\kappa_c$ approximation closely matches the angular standard deviation for $\kappa_c > 4$, which corresponds to $\sigma < 29°$. The angular standard deviation $\sigma = 55°$ reported by *Sargolini et al., 2006* corresponds to a concentration parameter $\kappa_c$ of about 1.7 and (in our large-$\kappa_c$ approximation) to a tuning width $1/\sqrt{\kappa_c} = 44°$ (upward triangle in *Figure 2—figure supplement 1B and C*).

### Numerical simulations

All simulations were implemented in Python 3.10 using the packages NumPy, SciPy, Numba, and JobLib. Matplotlib was used for plotting. Inkscape was used for the final adjustment to the figures.

## Acknowledgements

We would like to thank Tiziano D'Albis for discussions.

This work was funded by the German Research Foundation (DFG; project nos. 327654276 - SFB 1315 to RK), the German Federal Ministry of Education and Research (01GQ1705 to RK), and the Einstein Foundation Berlin (to IK). LK received funding from the German Research Foundation (DFG;

project nos. 447634521 and 527084865), the return program of the Ministry of Culture and Science of the State of North Rhine-Westphalia, the Federal Ministry of Education and Research (BMBF; 01GQ1705A), and by NIH/NINDS grant U01 NS113198-01.

## Additional information

### Funding

| Funder | Grant reference number | Author |
|---|---|---|
| Deutsche Forschungsgemeinschaft | 327654276 - SFB 1315 | Richard Kempter |
| Bundesministerium für Bildung und Forschung | 01GQ1705 | Richard Kempter |
| Einstein Center for Neurosciences Berlin | | Ikhwan Bin Khalid |
| Bundesministerium für Bildung und Forschung | 01GQ1705A | Lukas Kunz |
| National Institutes of Health | U01 NS113198-01 | Lukas Kunz |
| German Research Foundation | 447634521 | Lukas Kunz |
| German Research Foundation | 527084865 | Lukas Kunz |

The funders had no role in study design, data collection and interpretation, or the decision to submit the work for publication.

### Author contributions

Ikhwan Bin Khalid, Data curation, Software, Formal analysis, Validation, Investigation, Visualization, Methodology, Writing – original draft, Writing – review and editing; Eric T Reifenstein, Conceptualization, Data curation, Software, Formal analysis, Supervision, Validation, Investigation, Visualization, Methodology, Writing – original draft, Project administration, Writing – review and editing; Naomi Auer, Software, Formal analysis, Validation, Investigation, Visualization, Methodology, Writing – original draft, Writing – review and editing; Lukas Kunz, Conceptualization, Resources, Supervision, Funding acquisition, Visualization, Methodology, Writing – original draft, Project administration, Writing – review and editing; Richard Kempter, Conceptualization, Resources, Supervision, Funding acquisition, Methodology, Writing – original draft, Project administration, Writing – review and editing

### Author ORCIDs

Ikhwan Bin Khalid ⓘ https://orcid.org/0000-0003-1783-2834
Eric T Reifenstein ⓘ http://orcid.org/0000-0002-6898-0178
Naomi Auer ⓘ https://orcid.org/0000-0002-3978-8531
Lukas Kunz ⓘ https://orcid.org/0000-0002-0665-7703
Richard Kempter ⓘ https://orcid.org/0000-0002-5344-2983

### Decision letter and Author response

Decision letter https://doi.org/10.7554/eLife.85742.sa1
Author response https://doi.org/10.7554/eLife.85742.sa2

## Additional files

### Supplementary files

• MDAR checklist

### Data availability

The code used to generate the data is available at https://github.com/ikhwankhalid/grid_bold, copy archived at *Khalid, 2024*.

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
