## [Editor Report]

This computational work represents a valuable and long overdue assessment of the potential mechanisms associating patterns of activity of entorhinal grid cells, recorded mostly in rodents, with the population property of hexasymmetry detected in non-invasive human studies. The methodic comparison of alternative hypotheses is compelling, and the conclusions are important for the future design of experiments assessing the neural correlates of human navigation across physical, virtual, or conceptual spaces.

---

## [Decision Letter]

**Decision letter after peer review:**

Thank you for submitting your article "Quantitative modeling of the emergence of macroscopic grid-like representations" for consideration by *eLife*. Your article has been reviewed by 3 peer reviewers, and the evaluation has been overseen by a Reviewing Editor and Laura Colgin as the Senior Editor. The following individuals involved in the review of your submission have agreed to reveal their identity: Alessandro Treves (Reviewer #1); Daniel Bush (Reviewer #2); Matthew F Nolan (Reviewer #3).

Essential revisions (for the authors):

1) Strengthen the link between the simulations presented here and the known biophysical properties of rodent entorhinal grid cells. Revise the ranges used in the analyses in comparison with those reported in the literature (see details for each scenario below) and provide a graphical representation of the values in each case. Provide an idea of how values of H are related to results obtained with other methods in previous experimental studies.

2) Extend the study to null-hypothesis scenarios, such as the standard grid cell model or head-direction clustering that is independent of grid cell firing.

*Reviewer #1 (Recommendations for the authors):*

Suggestions for improvement to the authors:

Is there quantitative evidence independent of the Doeller (2010) paper that head direction selectivity is clustered along the grid axes?

Similarly, is there quantitative evidence independent of the Heys (2014) and Gu (2018) papers for the clustering of spatial phases?

The most important panels of the whole paper, Figure 2E, and Figure 3B are impossible to read. Could you replace them with level contours or proper color-coded ones?

Also, the gray curves in subplots Figure 2B-D and F-H are difficult to see.

Some statements seem to be overkill, such as that the adaptation weight should be restricted to the range 0 < w < 1.

Also, the detailed analysis of the third hypothesis seems like overkill.

For the navigation model, real rat trajectories are available; maybe human VR ones as well?

Are all grid units in the model on the same scale? The presence in rodent mEC of 4-5 "modules" with different grid scales should be discussed.

Can effective firing rate adaptation parameters be somehow gauged from studies of the real biophysical properties of grid cells, as in the early Giocomo and Hasselmo studies? Obtaining "realistic" parameters just by dividing by 2 ideal parameters seems a bit arbitrary.

Congratulations on the nice study, especially the valuable analytical components!

*Reviewer #2 (Recommendations for the authors):*

My main suggestion for improving this manuscript is to strengthen the link between the simulations presented here and the known properties of rodent entorhinal grid cells by: [1] clarifying exactly what each set of 'biologically plausible parameters' are and how they were obtained; and [2] providing graphical illustrations of these parameter values, to compare with the 'ideal conditions' that are displayed in most figures. Specifically:

In the conjunctive cell hypothesis section, could the authors describe how the 'biologically plausible' tuning width parameter was extracted from the data presented by Doeller et al. (Nature, 2010)? Towards the end of page 12, for example, the authors state that they "…derived the more realistic [tuning width] parameters from a previous study", but do not provide any details of these parameter values (I eventually found them in the caption of Figure 5) or how they were derived. Could they also add an illustration of the head direction tuning generated by all three tuning width parameters (i.e. those shown in panels F, G, and H) to Figure 2A, overlaid in dashed or dotted lines, to give an intuition of how each tuning width compares to empirical data?

In Figure 3, could the authors show the firing pattern of a simulated grid cell with the adaptation weight of 1 that is used to generate the results shown in panels C, D, and E? Does this introduce significant variability in firing rate between different firing fields, as observed experimentally (Ismakov et al., Current Biology, 2017)? What value of the adaptation weight was used to generate the data shown in dashed grey and pink lines on the right-hand side of panel A? Can this be related to estimates of the strength of firing rate adaptation in entorhinal pyramidal and stellate cells, for example as described by Alonso and Klink (Journal of Neurophysiology, 1993)? This might also lead the authors to modify their claim that they are "not aware of empirical investigations regarding repetition suppression effects in single grid cells", made at the top of page 13 and again in the Discussion

With regards to the structure-function hypothesis, and following the comments above, the authors should also state how the 'realistic' values of the clustering parameter that they refer to in Sections 2.2.3 and 2.3 were derived from data presented by Gu et al. (2018), and what those parameter values are

*Reviewer #3 (Recommendations for the authors):*

1. Could the null hypothesis of trajectory-dependent hexasymmetry be mapped out more clearly? For example:

- It's implicit that a 'standard' grid cell model wouldn't generate hexadirectional modulation. It could help the reader to make this clear early in the introduction.

- The Methods and Supplemental Data illustrate the possibility that hexadirectional modulation could arise simply from the trajectory followed – called path hexasymmetry here. The very nice modelling approach used provides an opportunity to quantify the size of this effect relative to effects in previous human studies and to verify that previous attempts to mitigate this through binning were successful. Adding this analysis would be a valuable control for the interpretation of human data.

- For the conjunctive grid model, is there anything special about the neurons being grid cells? In other words, would head direction cells with similar tuning profiles generate the same hexadirectional modulation? I expect they would but clarity here would be helpful for future mechanistic work.

2. For the evaluation of the grid by head direction hypothesis, the range of parameter values considered seems quite narrow compared to the likely properties of grid cells (see e.g. Sargolini et al. 2006). It could be helpful to add a null case where the head direction is not aligned to the grid axes and to consider a greater range of alignment jitter.

It could also help the reader to illustrate the head direction parameters for each simulation, alongside the results. E.g. With plots of the head direction tunings for the models presented.

3. The math behind the definition of H is clearly laid out in the methods section. I think it would help the reader to also give some intuition for the design and behaviour of H when it's introduced in the Results section. It would also be helpful to indicate here the range of possible values, and within that range to indicate values that could arise from noisy activity given finite sampling duration, etc.

4. Related to the above, some intuition for the expected values of H given previous fMRI/EEG studies could also be helpful. For example, to facilitate comparisons could the numerical simulation results also be analysed with previous methods and the scores reported? This way a reader could better appreciate the extent to which the effect sizes in the simulations are comparable with the human fMRI/EEG data.

---

## [Author Response]

Essential revisions (for the authors):1) Strengthen the link between the simulations presented here and the known biophysical properties of rodent entorhinal grid cells. Revise the ranges used in the analyses in comparison with those reported in the literature (see details for each scenario below) and provide a graphical representation of the values in each case. Provide an idea of how values of H are related to results obtained with other methods in previous experimental studies.

Thank you for these suggestions. In the revised manuscript, we have taken the following actions to address this comment. In brief:

We have revisited the known biophysical properties of rodent entorhinal grid cells and have updated the ranges of these properties in our analyses. We have recomputed our simulations using the updated values and have adjusted our figures (e.g., Figure 2 and Figure 5).We now provide more graphical representations of the grid-cell parameters that we used for our simulations (e.g., Figure 2A).We now provide a detailed comparison of our new *H* metric to the outcomes of previously used methods for estimating hexadirectionality in neural signals (Figure 5 —figure supplement 4).

Details are explained in our responses to the comments by the reviewers.

2) Extend the study to null-hypothesis scenarios, such as the standard grid cell model or head-direction clustering that is independent of grid cell firing.

We have revised the manuscript in several ways to address this comment by describing null-hypothesis scenarios more clearly. In brief:

Regarding the standard grid-cell model, we now describe that it is a special case of the structure-function mapping hypothesis in which we randomly draw the grid phase offsets from a uniform distribution on the unit rhombus. This scenario leads to significant hexasymmetry due to sampling a finite number of grid phase offsets (Figures. 4, 5, and Figure 4 —figure supplement 1).Regarding the head-direction clustering, we now explain in the manuscript (e.g. in the Introduction) that head-direction clustering independent of grid-cell firing is a special case of the conjunctive hypothesis in which the preferred firing directions of the head-direction cells are not aligned with the grid axes. This special case can lead to hexasymmetry values that are as strong as for the case in which the preferred firing directions are aligned with the grid axes.

Again, details are explained in our responses to the comments by the reviewers.

Reviewer #1 (Recommendations for the authors):Suggestions for improvement to the authors:1. Is there quantitative evidence independent of the Doeller (2010) paper that head direction selectivity is clustered along the grid axes?

We are not aware of other studies that have replicated this finding and we have added this notion to the manuscript (fourth paragraph of the Introduction): “… though further studies are needed to corroborate this observation.” There is in fact one paper (Keinath, 2016) that reports 10-fold symmetry in entorhinal grid cells instead of 6-fold symmetry. We now reference this paper in our manuscript (second paragraph of section “A note on our choice of the values of model parameters” in the Discussion): “… the preferred head directions of conjunctive grid-by-head-direction cells exhibit other types of rotational symmetry such as 10-fold rotational symmetry reported in recordings from rats (Keinath, 2016).”

2. Similarly, is there quantitative evidence independent of the Heys (2014) and Gu (2018) papers for the clustering of spatial phases?

We have revisited the literature on this topic and while we found evidence for the topographic arrangement of spatially modulated neurons in the medial entorhinal cortex (Obenhaus et al., PNAS, 2021; Naumann et al., Journal of Neurophysiology, 2018) (now cited in the 6th paragraph of the Introduction), we did not find more evidence for a clustering of grid phases.

3. The most important panels of the whole paper, Figure 2E, and Figure 3B are impossible to read. Could you replace them with level contours or proper color-coded ones?

Following your suggestion, we have added level contours to both subfigures to improve legibility.

4. Also, the gray curves in subplots Figure 2B-D and F-H are difficult to see.

We have changed the gray curves to magenta dashed curves and brought them to the foreground for visibility.

5. Some statements seem to be overkill, such as that the adaptation weight should be restricted to the range 0 < w < 1.

We have considerably changed the text in the Results (section “Repetition suppression hypothesis”), but we have kept “0 < *w_r_* < 1” because we think it is relevant information. *w_r_* > 0 is needed because otherwise there is no adaptation. *w_r_* < 1 ensures that adaptation is not too strong (by Equation 5, *w_r_* < 1 ensures a finite firing rate G_i_^r^ > 0 in case of a constant input G_i_ > 0). Hence, in the Results we have shortened the respective paragraph including the following change: we have replaced “… as negative values would cause enhancement rather than suppression, and values larger than one would lead to suppression that is stronger than the peak activity of the single grid cell (Figure 3B).” by “… the full range of reasonable values (0 < *w_r_* < 1)…”.

6. Also, the detailed analysis of the third hypothesis seems like overkill.

Thanks for the suggestion to simplify the related Figure 4; panels G-T are indeed complex. We believe, however, that the higher-order spatial clustering is a relevant property of the clustering hypothesis and that it deserves some attention. As we show in Figure 4S, higher-order spatial clustering leads to higher values for the clustering concentration parameter (κs) and, thus, to higher neural hexasymmetry. Furthermore, Figures 4R and 4T are essential for an overall evaluation of the structure-function mapping hypothesis. We would thus prefer to keep the figure as is.

7. For the navigation model, real rat trajectories are available; maybe human VR ones as well?

We have now performed our simulations on real rat trajectories and real trajectories from human virtual-reality experiments (Kunz et al., 2015; Kunz et al., 2021). We find that the navigation patterns of both rats and humans lead to significant neural hexasymmetry (as compared to path hexasymmetry) only in the case of the conjunctive hypothesis (new supplementary Figure 5 —figure supplement 3). This is due to the fact that both rat and human navigation paths are too curved to allow for repetition suppression or structure-function mapping to take effect. Although the human navigation paths are straighter than those of rats, they are still closer related to random walks than to piecewise-linear walks. We now report this observation in the main text (second paragraph of section “Navigation paths have a major influence on the hexadirectional population signal” in the Discussion): “We note that in our simulations, each linear path segment of a piecewise linear walk has a length that is ten times larger than the grid scale. While humans performing navigation tasks exhibit straighter trajectories than those of rats (Doeller et al., 2010; Kunz et al., 2015; Horner et al., 2016), they are still more similar in scale to random walks than piecewise linear walks. Thus, when using empirical navigation paths from previous studies, neural hexasymmetries were significant only for the conjunctive hypothesis (Figure 5 —figure supplement 3)”.

We note though that there is a variety of different virtual-reality navigation tasks in humans with varying degrees of straightness. For example, in Horner et al., Current Biology, 2016, the navigation paths very clearly resemble piecewise-linear walks. In such studies, repetition-suppression effects may play a role in the emergence of grid-like representations.

8. Are all grid units in the model on the same scale? The presence in rodent mEC of 4-5 "modules" with different grid scales should be discussed.

Thanks for this suggestion. Following your comment, we state in the Discussion (first sentence of section “Influence of grid scale on hexasymmetry”) and in section “Implementation of the three hypotheses to explain macroscopic grid-like representations” of the Methods that all grid cells in our simulations have the same grid spacing of 30 cm.

We now explicitly describe that rodent grid cells are organized in four to five modules (beginning of paragraph 2 of section “Influence of grid scale on hexasymmetry” of the Discussion): “Another factor that might complicate the picture is an interaction of different grid scales: rodent grid cells have been found to be organized into 4-5 discrete modules with different grid scales (Stensola et al., 2012) and this may also be true in humans. Together with the anatomical dorsoventral length of the human entorhinal cortex (≈ 2 cm, Behuet et al., 2021), this means that a module might cover roughly 4-5 mm. A typical voxel of 3x3x3 mm³ would thus either contain one or two grid modules. The case of a voxel containing a single module was already discussed. If a voxel contained two modules, there would be two subpopulations of grid cells with different grid spacings.”

We have also added new text to the Discussion to explain the implications of different modules for the three hypotheses (end of paragraph 2 of section “Influence of grid scale on hexasymmetry”): “For the structure-function mapping hypothesis, significant neural hexasymmetry can be achieved only for star-like trajectories. In this scenario, the hexasymmetry depends on the starting position of the trajectories, i.e., the center position of the star (Figure 4D). Two subpopulations of grid cells from different modules essentially act like two individual grid cells in the context of the structure-function mapping hypothesis. Thus, they can either add their hexasymmetry values (e.g. when the star center is in the middle of a grid field for both subpopulations) or suppress each other (when the star center is in a regime of 0° grid phase for one subpopulation and in a regime of ± 30^o^ grid phase for the other). For the repetition suppression hypothesis, two populations of grid cells with the same adaptation time constant but different grid spacing would result in two different values of hexasymmetry, depending on the interaction between grid-field traversal time and adaptation time constant. The joint hexasymmetry value would then depend on the two subpopulation sizes. We note that the adaptation time constant might be tuned to match the grid spacing. This would result in high hexasymmetry values for both subpopulations, and thus also for the full population.”

We now also discuss the effect of changing the grid scale (first paragraph of section “Influence of grid scale on hexasymmetry” of the Discussion): “In this study, we assumed that all grid cells have the same grid spacing of 30 cm. For rats, this is at the lower end of grid spacing values, corresponding to the dorsal region of the medial EC (Stensola et al., 2012). How would the three hypotheses perform for larger values of grid spacing? The conjunctive hypothesis would remain unchanged as it does not depend on grid spacing. In contrast, the repetition suppression hypothesis depends on the grid scale. Here, the strongest hexasymmetry is achieved when the adaptation time constant is similar to the average time that it takes the subject to move between neighboring grid fields (Figure 3B). Altering the grid scale changes this traversal time between grid fields and thus hexasymmetry. In Figure 3, we chose the "ideal" adaptation time constant to maximize hexasymmetry. A change in grid scale would thus reduce hexasymmetry for this particular value of the adaptation time constant. As there are different grid-cell modules in the entorhinal cortex that exhibit different grid spacings (Stensola et al., 2012), repetition-suppression effects may vary across modules (if the adaptation time constant is not tuned in accordance with the grid scale). Regarding the structure-function mapping hypothesis and star-like walks, different grid spacings can have a major effect on neural hexasymmetry if the path segments are short compared to the grid scale, similar to effects of the starting location (Figure 4D); for longer path segments, altering the grid scale should have a smaller effect on neural hexasymmetry. For the structure-function mapping hypothesis, random walks and piecewise-linear walks do not lead to real neural hexasymmetries, and different grid spacings are thus irrelevant. Overall, different grid spacings are most important in the context of the repetition suppression hypothesis.”

9. Can effective firing rate adaptation parameters be somehow gauged from studies of the real biophysical properties of grid cells, as in the early Giocomo and Hasselmo studies? Obtaining "realistic" parameters just by dividing by 2 ideal parameters seems a bit arbitrary.

We agree with you. To find an empirical quantification of the adaptation parameters, we have looked through a series of studies on resonant properties and time constants in rat entorhinal principal cells (Alonso and Klink, 1993; Giocomo and Hasselmo, 2008; Giocomo and Hasselmo, 2009). They report adaptation with time constants of <100 ms, whereas the repetition-suppression hypothesis in our study requires adaptation time constants of a few seconds (given realistic values for grid distance and the subject’s speed).

There are examples of slower adaptation time constants: Figure 2F in Giocomo and Hasselmo, (2008) shows that, in ventral mEC, the slow component of the hyperpolarization-activated current (Ih) has time constants of 2-3 s. A second example is Magistretti and Alonso (1999) who observed second-long inactivations of a persistent sodium current in EC layer II cells.

Furthermore, Reber et al. (2023, their figure 2B, right) quantified adaptation in human single neurons. In that study, adaptation strengths of concept cells were about 5% in average firing rates in the hippocampus, entorhinal, and parahippocampal cortex for stimuli separated by several seconds. In our work, we used an adaptation strength (ideal parameters) of ~20% over a similar timescale (Figure 3A, right). Even for realistic parameters, the adaptation strength (~16%) is larger but of the same order of magnitude as the attenuation found by Reber et al. (2023). We thus believe that our current value for “realistic” adaptation strengths is fine, and we have thus kept this value for our simulations.

We added all these points to our new Methods section “Parameter Estimation”.

Congratulations on the nice study, especially the valuable analytical components!

Thank you!

Reviewer #2 (Recommendations for the authors):My main suggestion for improving this manuscript is to strengthen the link between the simulations presented here and the known properties of rodent entorhinal grid cells by: [1] clarifying exactly what each set of 'biologically plausible parameters' are and how they were obtained; and [2] providing graphical illustrations of these parameter values, to compare with the 'ideal conditions' that are displayed in most figures. Specifically:In the conjunctive cell hypothesis section, could the authors describe how the 'biologically plausible' tuning width parameter was extracted from the data presented by Doeller et al. (Nature, 2010)? Towards the end of page 12, for example, the authors state that they "…derived the more realistic [tuning width] parameters from a previous study", but do not provide any details of these parameter values (I eventually found them in the caption of Figure 5) or how they were derived. Could they also add an illustration of the head direction tuning generated by all three tuning width parameters (i.e. those shown in panels F, G, and H) to Figure 2A, overlaid in dashed or dotted lines, to give an intuition of how each tuning width compares to empirical data?

Thanks for these helpful suggestions. We now describe in our new Methods section “Parameter Estimation” how the biologically plausible parameters were derived. Specifically, for the conjunctive cell hypothesis, we explain in detail, for example, how the tuning width parameter was extracted from Supplementary Figure 3 of Doeller et al. (2010).

Following your comment, we have added illustrations of the head-direction tuning generated by the parameter combinations for panels F, G, and H in Figure 2. We have also modified Figure 2A, right, to more clearly show the jitter, tuning width, and the resulting effective head-direction tuning for a single conjunctive grid cell.

In Figure 3, could the authors show the firing pattern of a simulated grid cell with the adaptation weight of 1 that is used to generate the results shown in panels C, D, and E? Does this introduce significant variability in firing rate between different firing fields, as observed experimentally (Ismakov et al., Current Biology, 2017)? What value of the adaptation weight was used to generate the data shown in dashed grey and pink lines on the right-hand side of panel A? Can this be related to estimates of the strength of firing rate adaptation in entorhinal pyramidal and stellate cells, for example as described by Alonso and Klink (Journal of Neurophysiology, 1993)? This might also lead the authors to modify their claim that they are "not aware of empirical investigations regarding repetition suppression effects in single grid cells", made at the top of page 13 and again in the Discussion

We have divided our response into 3 parts:

(1) Following your suggestions, we now show the firing pattern of a simulated grid cell with the adaptation weight of 1 in Figure 3A, indicating a reduction in firing rate along a “misaligned” path and a stronger reduction for an “aligned” path. In the caption of Figure 3, we have added the information which adaptation weight was used to produce Figure 3A,C,D,E: “adaptation weight *w*_r_ = 1 and time constant τ_r_ = 3 s”. In all panels (apart from the overview in Figure 3B), we have used the same parameters.

(2) Regarding your question whether repetition suppression may lead to the findings by Ismakov et al., 2017, we performed new simulations to examine whether firing-rate adaptation in single trials can account for the effects observed by Ismakov et al., 2017. We simulated 1024 grid cells with an adaptation weight *w*_r_ = 1 and a time constant of τ_r_ = 3 s and random-walk trajectories (which resemble those of rodents; with a duration between 10-10,000 seconds) in a finite environment, and quantified the variability of the peak firing rates across grid fields using the coefficient of variation, separately for each cell (Author response image 1). This procedure is the same as in Ismakov et al., 2017.

Our new simulations show that peak firing rate is indeed variable, and that repetition suppression mainly reduces the peak firing rate (Author response image 1). In principle, both grid cells with and without repetition suppression can exhibit variations in peak firing rates across grid fields, and these variations can resemble those seen by Ismakov et al., 2017 (Author response image 1). Specifically, for a simulation duration of about 300-400 seconds, the coefficient of variation is about 0.6 (Author response image 1), similar to the values reported by Ismakov et al., 2017 (their Figure 1B, CV ≈ 0.58). These variations are driven by grid-field traversals that do not cross the exact center of the grid field, incomplete grid fields at boundaries, and repetition suppression effects in the case of grid cells that are modeled with repetition suppression.

Our simulations also show that the duration of the simulation has a major effect on the coefficient of variation: short durations (<200 seconds) lead to higher coefficients of variations than those reported by Ismakov et al., 2017, and long durations (>1,000 seconds) lead to lower coefficients of variation. Furthermore, grid cells outfitted with repetition suppression generally exhibit higher coefficients of variation, and this effect is more pronounced for shorter simulation durations (Author response image 1). For very long simulation durations, this difference disappears, and repetition suppression does not have a significant effect on the variability of peak firing rates between grid fields, despite an overall reduction in peak firing rates (Author response image 1). Taken together, repetition suppression may account for the findings by Ismakov et al., 2017, but this depends on the duration of the recording. Furthermore, repetition suppression is not strictly needed to produce the observed variability in peak firing rates.

**Author response image 1. sa2fig1:** Variability of peak firing rates of grid cells when using a random walk trajectory in a finite environment. (A) Left: The distribution of the peak firing rates for the two conditions (repetition-suppression hypothesis: orange; without repetition-suppression: blue) for 1024 grid cells across 300 simulations and a simulated duration of 9000 seconds, which is kept constant for all panels of (A). Right: The distribution for the coefficient of variation (CV) of the peak firing rates across all cells. The dashed lines represent distribution means. (B) The dependence of the mean CV on the simulated duration for 1024 grid cells across 300 simulations. As the simulated duration increases, the difference in CV between the two cases decreases. The shaded area represents the standard error. (C) Four examples of the rate map for different values of the simulated duration. In the rightmost panel, the trajectory is not shown so that the rate map can be seen clearly. We employed the 'ideal’ parameter set for repetition suppression: τr = 3 s, wr = 1; in all simulations, cells did not show any head direction-tuning, and there was no clustering of grid-phase offsets. The size of the environment was limited to a square with side lengths of 100 cm. Mann-Whitney U test; ***, *P* < 0.001; **, *P* < 0.01; *, *P* < 0.05; n.s., not significant.

(3) Finally, we turn to the question about the strength of firing-rate adaptation in entorhinal pyramidal and stellate cells (see also Reviewer #1, recommendation #9:). We have summarized the current state of the literature in the new Methods section “Parameter estimation”. Regarding Alonso and Klink (1993): they report adaptation with time constants of <100 ms; furthermore, their results were obtained in vitro, and they did not know whether the cells that they characterized were grid cells. We would thus like to keep our statement (in the Discussion) that we are currently not aware of detailed measurements of the relevant grid cell properties.

With regards to the structure-function hypothesis, and following the comments above, the authors should also state how the 'realistic' values of the clustering parameter that they refer to in Sections 2.2.3 and 2.3 were derived from data presented by Gu et al. (2018), and what those parameter values are

We have added this information to our new Methods section “Parameter estimation”. In brief, we performed random-field simulations with different spatial correlation lengths and compared the results to the experimental findings by Gu et al. (2018) and Heys et al. (2014). The ‘realistic’ values for the clustering parameters were then derived from the simulations that reflected the experimental data most closely.

Reviewer #3 (Recommendations for the authors):1. Could the null hypothesis of trajectory-dependent hexasymmetry be mapped out more clearly? For example:- It's implicit that a 'standard' grid cell model wouldn't generate hexadirectional modulation. It could help the reader to make this clear early in the introduction.

We assume that the reviewer uses the term ‘standard’ grid cell model to refer to a grid-cell population with phase offsets randomly drawn from a uniform distribution on the unit rhombus. Our simulations show that a realization of such a grid-cell population nevertheless has a nonzero sample mean clustering concentration parameter κ^s>0, the value of which depends on the (finite) number of grid cells (Figure 4R). This leads to some small hexasymmetry (H = 0.7 spk/s; Figure 4 —figure supplement 1, “standard, star”) that is significantly larger than the path hexasymmetry. Hence, we do not want to refute the ‘standard’ grid cell model already in the Introduction. Note, however, that distributing the grid phases regularly throughout the unit rhombus (and not randomly as before) results in an estimated concentration parameter κ^s=0. In this case there is indeed no clustering. In additional simulations we found that the resulting neural hexasymmetry was negligible (H <10−10 spk/s).

We have added this information as a new supplementary figure (Figure 4 —figure supplement 1). In the Discussion, we extend on this by describing the relatively low hexasymmetry values for the structure-function mapping hypothesis for clustering concentration parameters that are low or even zero (third paragraph of section “A note on our choice of the values of model parameters”): “Specifically, we found that the clustering concentration parameters of grid phase offsets gleaned from existing empirical studies (Gu et al., 2018) produced the smallest neural hexasymmetry when compared to the other conditions we considered, while the neural hexasymmetries obtained from simulations randomly sampling grid phase offsets from a uniform distribution on the unit rhombus (that one might refer to as the ‘standard grid cell model’) were only slightly smaller.” We mention these results also in the caption of Figure 5.

- The Methods and Supplemental Data illustrate the possibility that hexadirectional modulation could arise simply from the trajectory followed – called path hexasymmetry here. The very nice modelling approach used provides an opportunity to quantify the size of this effect relative to effects in previous human studies and to verify that previous attempts to mitigate this through binning were successful. Adding this analysis would be a valuable control for the interpretation of human data.

We performed extensive simulations to compare our new method to previous ways of quantifying hexasymmetry in neural data. This shows qualitatively identical results for all different approaches (Figure 5 —figure supplement 4), indicating that the analytical approach of previous studies was valid (for details, see our response to your comment #1 in the Public Review)*.*

- For the conjunctive grid model, is there anything special about the neurons being grid cells? In other words, would head direction cells with similar tuning profiles generate the same hexadirectional modulation? I expect they would but clarity here would be helpful for future mechanistic work.

The conjunctive grid by head-direction cell hypothesis does not necessitate that the cells are grid cells. Moreover, in the case that they are grid cells, the head direction does not need to be aligned to the grid axes (see also your recommendation #4 below). We have clarified this in the revised Introduction (end of fourth paragraph of the Introduction) and return to it in the Discussion (section “Potential mechanisms underlying hexadirectional population signals in the entorhinal cortex”).

2. For the evaluation of the grid by head direction hypothesis, the range of parameter values considered seems quite narrow compared to the likely properties of grid cells (see e.g. Sargolini et al. 2006). It could be helpful to add a null case where the head direction is not aligned to the grid axes and to consider a greater range of alignment jitter.

We agree with the reviewer and have thus revised Figure 2E to show neural hexasymmetries for tuning widths up to 40 degrees and jitters up to 30 degrees. Additionally, we added a new supplementary figure (Figure 2 —figure supplement 1) where we extended the range of tuning widths to 60 degrees. Figure 2 —figure supplement 1C shows the strong dependence of hexasymmetry on tuning width and jitter. We found significant hexadirectional modulation for a tuning width of about < 30^o^ (for an alignment jitter of about < 15°) with a steep increase of hexasymmetry for sharper tuning. On the other hand, hexasymmetry appears to saturate at noise levels (H ~ 2 spikes/s) for tuning widths larger than about 35^o^.

Doeller et al. (2010, their Supplementary Figure 3) reported that the tuning curves usually cover between one sixth and one third of a circle, corresponding to tuning widths between roughly 30^o^ and 60^o^. Sargolini et al. (2006) reported values that correspond to tuning widths of about 44^o^ (upward triangle in Figure 2 —figure supplement 1C); note that Doeller et al. and Sargolini et al. had different criteria for identifying HD cells, which could account for different tuning widths in these studies. Surprisingly, the average tuning widths reported in these empirical studies are quite large (>35^o^) and thus would lead to hexasymmetry values that are basically at the noise level (H ~ 2 spikes/s).

However, it should be emphasized that (unlike in our simulations) the tuning widths reported by these empirical studies are population averages that might consist of a large number of widely tuned cells and a small subpopulation of narrowly tuned cells. In this case, the narrowly tuned subpopulation would still give rise to a strongly hexadirectionally modulated signal under the framework of the conjunctive grid by head-direction cell hypothesis. We thus chose the tuning width parameter to correspond to the lower end of the reported range in Doeller et al. (2010), i.e., we set (in the revised manuscript) the tuning width to 29^o^ (roughly one sixth of a circle), which corresponds to a value of κc=4 for the concentration parameter for direction tuning.

Finally, we note that our definition for the tuning width is different to that reported by Sargolini et al. We added new text to explain this difference (Methods section “Tuning widths and the large-κc approximation”): “The numerical values of our tuning width of head-direction modulated cells should not be directly compared to the numbers reported by Sargolini et al. (2006) with values in the range of 55^o^ because Sargolini et al. reported the tuning widths as the angular standard deviation σ of the mean vector of head-direction dependent firing rates whereas we derive the tuning width from the large-κc approximation of a von Mises distribution: σ=1/κc. This simple approximation is reasonably good only for large enough κc (or small enough σ), and 55^o^ is already beyond the range where the approximation is reasonable (Figure 2 —figure supplement 1B).

To examine the range of values where the large-κc approximation holds, we can compare it to the angular standard deviation expressed as σ=−2, ln[ I1(κc)/I0(κc)] where In(κc) is the modified Bessel function of the first kind of order ‘n’. This equation is valid for all values of the concentration parameter κc. The results of this comparison are summarized in Figure 2 —figure supplement 1B. In brief, the large-κc approximation closely matches the angular standard deviation for κc>4, which corresponds to σ < 29^o^. The angular standard deviation σ = 55^o^ reported by Sargolini et al. corresponds to a concentration parameter κc of about 1.7 and (in our large-κc approximation) to a tuning width 1/κc= 44^o^ (upward triangle in Figure 2 —figure supplement 1B and C).”

It could also help the reader to illustrate the head direction parameters for each simulation, alongside the results. E.g. With plots of the head direction tunings for the models presented.

We have added illustrations of the head-direction tuning generated by the parameter combinations for panels F, G, and H in Figure 2. We have also modified Figure 2A, right, to more clearly show the jitter, tuning width, and the resulting effective head-direction tuning for a single conjunctive grid cell.

3. The math behind the definition of H is clearly laid out in the methods section. I think it would help the reader to also give some intuition for the design and behaviour of H when it's introduced in the Results section. It would also be helpful to indicate here the range of possible values, and within that range to indicate values that could arise from noisy activity given finite sampling duration, etc.

We have now included a statement in the Methods section regarding the range of possible values for the hexasymmetry: “The hexasymmetry is nonnegative (H>=0) and it has the unit spk/s (spikes per second).” (section “Quantification of hexasymmetry of neural activity and trajectories”, right below Equation 12). We have also added this information to the Results (section “Quantifying neural hexasymmetry generated by the three hypotheses”), along with two examples for different hexasymmetry values: “The hexasymmetry *H* has a value of *H*=0 if there is no hexadirectional modulation, e.g., if the population firing rate does not depend on the movement direction. Conversely, if the population firing rate has a hexadirectional sinusoidal modulation, half of its amplitude (in units of the firing rate) equals the value of the hexasymmetry H. Using the same approach, we quantified the hexasymmetry of a trajectory and called this path hexasymmetry (see Methods after equation 14).”

Following your suggestion to indicate the values that could arise from noisy activity, we have added this information to the manuscript (Discussion, new section “Sources of noise”): “The neural hexasymmetry as defined by Equation (12) also contains contributions from the path hexasymmetry of the underlying trajectory. Star-like walks and piecewise linear walks have a path hexasymmetry of zero by construction. Therefore, these trajectory types do not contribute to the neural hexasymmetry. For random walks, the path hexasymmetry depends on the number of time steps M and the tortuosity parameter σ_θ_ (Figure 1 —figure supplement 4). To compare the path hexasymmetry to the neural hexasymmetry, the path hexasymmetry needs to be multiplied by the time-averaged population activity. Since the path hexasymmetry ranges from zero to one (Equation 14), its contribution to the neural hexasymmetry ranges from zero (for a uniform sampling of movement directions) to the time averaged population activity. For M = 900 000 time steps and a tortuosity of σ_θ_ = 0.5 rad/s^1/2^, random walk trajectories contribute ~0.8% of the time-averaged population activity to the neural hexasymmetry (Figure 1 —figure supplement 4). In our simulations with an average population activity in the range of 10^3^ spk/s, the 0.8% noise floor leads to a path hexasymmetry of about 8 spk/s (Figure 5).

Another factor that contributes noise is the finite sampling of grid phase offsets. For all our simulations of the conjunctive grid by head-direction cell hypothesis and the repetition suppression hypothesis (and for control simulations of the structure function mapping hypothesis), we sample a finite number of grid phase offsets from a two-dimensional von Mises distribution with a clustering concentration parameter κs=0 (equivalent to a uniform distribution on the unit rhombus). In this scenario, random fluctuations give rise to realizations with a non-zero sample mean clustering concentration parameter κ^s, which led to a corresponding mean neural hexasymmetry of 0.7 spk/s (labeled “standard” and “star” in Figure 4 —figure supplement 1), even in the absence of conjunctive tuning and repetition suppression. For 1024 grid cells, this contribution is ten times smaller than the neural hexasymmetry resulting from the ‘’realistic’’ parameters for both the conjunctive grid by head-direction cell hypothesis and the repetition suppression hypothesis (Figure 5). In the case of the structure-function mapping hypothesis, these fluctuations would not be considered as ‘noise’ as they directly contribute to the hexasymmetry from the clustering of grid phase offsets.”

For further comments about noisy activity, we also would like to refer to our response to *Reviewer #1, comment #3* and the new Figure 2 —figure supplement 2. The noise floor is also indicated in the new Figure 2 —figure supplement 1C.

4. Related to the above, some intuition for the expected values of H given previous fMRI/EEG studies could also be helpful. For example, to facilitate comparisons could the numerical simulation results also be analysed with previous methods and the scores reported? This way a reader could better appreciate the extent to which the effect sizes in the simulations are comparable with the human fMRI/EEG data.

Thanks for these suggestions. Since we do not model the transformation from population firing rates to fMRI/EEG signals, it is not possible for us to state explicit values of *H* for these previous studies. We predict that our hexasymmetry values pose an upper bound for hexadirectional modulation in fMRI/EEG experiments, since we do not expect the transformation from firing rates to macroscopic signals to introduce any additional hexasymmetry.

Nevertheless, following your suggestion, we have now used previous methods to analyze our numerical simulations. We compared the statistical outcomes from these previous methods to our statistical results. Importantly, the statistical results from the different methods are qualitatively identical for the different approaches (Figure 5 —figure supplement 4), demonstrating that all previous analytical approaches can be used to assess the significance of neural hexasymmetry.